# Room Temperature Chemiresistive Gas Sensors Based on 2D MXenes

**DOI:** 10.3390/s23218829

**Published:** 2023-10-30

**Authors:** Ali Mirzaei, Myoung Hoon Lee, Haniyeh Safaeian, Tae-Un Kim, Jin-Young Kim, Hyoun Woo Kim, Sang Sub Kim

**Affiliations:** 1Department of Materials Science and Engineering, Shiraz University of Technology, Shiraz 71557-13876, Iran; mirzaei@sutech.ac.ir (A.M.); haniyeh.safaeyan@gmail.com (H.S.); 2Department of Materials Science and Engineering, Inha University, Incheon 22212, Republic of Korea; dm249@naver.com (M.H.L.); xodjs635@naver.com (T.-U.K.); 3The Research Institute of Industrial Science, Hanyang University, Seoul 04763, Republic of Korea; piadote@naver.com; 4Division of Materials Science and Engineering, Hanyang University, Seoul 04763, Republic of Korea

**Keywords:** 2D nanomaterials, MXene, sensing mechanism, gas sensor, RT

## Abstract

Owing to their large surface area, two-dimensional (2D) semiconducting nanomaterials have been extensively studied for gas-sensing applications in recent years. In particular, the possibility of operating at room temperature (RT) is desirable for 2D gas sensors because it significantly reduces the power consumption of the sensing device. Furthermore, RT gas sensors are among the first choices for the development of flexible and wearable devices. In this review, we focus on the 2D MXenes used for the realization of RT gas sensors. Hence, pristine, doped, decorated, and composites of MXenes with other semiconductors for gas sensing are discussed. Two-dimensional MXene nanomaterials are discussed, with greater emphasis on the sensing mechanism. MXenes with the ability to work at RT have great potential for practical applications such as flexible and/or wearable gas sensors.

## 1. Introduction

Chemiresistive gas sensors, in which resistance is modulated in the presence of target gases, are fabricated mostly using metal oxide semiconductors (MOSs). This is not only owing to their high response, high stability, and fast dynamics, but also their low price and simple design [1,2]. Nevertheless, their working temperature is often high, up to 450 °C, leading to high power consumption. This is due to the fact that at low temperatures there is no sufficient energy for the gas molecules to overcome the adsorption barrier. However, with an increase in the sensing temperature, more gas molecules can be adsorbed on the surface of the sensor, leading to enhanced gas response. It should be noted that some metal oxides are able to work at RT [3]. However, some metal oxides have a very high resistance, in the range of Giga ohms, at RT, and some of them show too much noise and drift at RT, making difficult to measure their electrical properties. In contrast, MXenes have much lower resistance at RT with less noise, making them more favorable for sensing applications at RT. However, for some metal oxides which are able to work at RT, they generally can offer a higher response relative to MXene-based gas sensors due to their high intrinsic sensing properties.

Even though an increase of the sensing temperature is beneficial to have a higher response, it will result in significant power consumption. This is a disadvantage for gas sensors, especially for those working in remote areas, because of the limitation in power sources. Therefore, this issue limits the long-term use of high-temperature gas sensors in remote areas. Also, operation of the sensor at high temperatures for long times leads to a decrease in the sensor stability. Indeed, at high temperatures, more reactions occur on the sensor surface, resulting in generation of new phases or even poisoning of the sensor, which eventually may degrade the sensing properties. Furthermore, at high temperatures ultrafine particles of the sensing layer may gradually sinter together, leading to agglomerations of particles and degradation of the sensing properties. Moreover, the detection of explosive gases, such as H_2_ and CH_4_, at high temperatures is associated with an explosion risk. Hence, it is important to develop low- or room-temperature (RT) gas sensors [4].

In this regard, two-dimensional (2D) MXenes have gained more attention than other semiconductors for the realization of RT gas sensors because of their good conductivity, high surface area, ease of synthesis, tunable band gaps, ease of operation, and unique electrical characteristics. Furthermore, they have plenty of functional groups on their surfaces which are favorable sites for adsorption of gas molecules, which makes them very sensitive to changes in the gas concentrations in the surrounding atmosphere [5]. By controlling the type and amount of functional groups, it is possible to increase the sensitivity of these sensing materials. Owing to working at RT, they can be used for detection of explosive gases with high safety. In addition, they can be used in electronic devices and smartphones because of their low power consumption. Furthermore, they can be used for realization of flexible and wearable gas sensors which have applications in wearable electronic devices [6,7]. 

Previously, Li et al. [8] reviewed MXenes from chemical, electrochemical and energy-storage application points of view. Bhat et al. [9] discussed the stability of 2D MXenes for clean energy conversion and storage applications. Murali et al. [10] reviewed the synthesis of MXenes, their stability and photocatalytic applications. In another review paper, Murali and co-workers [11] discussed supercapacitor applications of MXenes. Also, recently, gas-sensing properties of MXenes have been reviewed. Ma et al. [7] reviewed flexible MXene-based gas sensors for wearable applications including gas sensors. Bhardwaj et al. reviewed MXene-based gas sensors [12]. Mehdi Aghaei and co-workers [13] discussed experimental and theoretical advances in MXene-based gas sensors. Zhang et al. [14] discussed strategies and challenges for improving performance of MXene-based gas sensors. Decaraj et al. discussed MXene and its nanocomposites for the detection of inorganic gases [15]. Xia et al. [16] discussed developments and challenges in MXene-based chemical gas sensors. Nahirniak and co-workers [17] reviewed MXene-heterojunctions for gas sensing application.

Riazi and co-workers [18] discussed MXene-based nanocomposites for gas sensing applications. Pei et al. [19] discussed gas sensing characteristics of Ti_3_C_2_T*_X_* MXene. Deshmukh et al. [20] Shina et al. [21] reviewed gas sensing and bio sensing properties of MXenes. Otgonbayar et al. discussed MXene sensors including gas sensors [22]. Ta et al. [23] discussed functional Ti_3_C_2_T*_x_* MXene for gas-sensing application. Jin et al. reviewed MXene-based textile sensors for wearable applications [24]. Radhakrishnan and co-workers [25] discussed MXenes for next-generation RT NO_2_ gas sensors. Peng et al. [26] discussed Ti_3_C_2_T*_x_*-based gas sensors. Gautam et al. [27] discussed MXene and graphene for different applications, including gas-sensing application. Lee et al. [28] discussed MXene gas sensors from both theoretical and experimental points of view. Xin et al. [29] discussed MXenes in wearable sensing devices. Sivasankarapillai et al. [30] reviewed MXene-based materials for sensing applications. Simonenko et al. [31] discussed the gas-sensing properties of Ti_2_C_2_T*_x_* MXene. Tran et al. [32] reviewed modified MXenes for sensing studies. Li et al. reviewed gas-sensing properties of 2D MXenes [33].

In this review, we have comprehensively reviewed the recent advancements in the development and application of RT chemiresistive gas sensors based on 2D MXenes, which have received less attention in previous review papers. In more detail, the objective of this review paper is to cover the synthesis methods, performance characteristics, sensing mechanism, and potential applications of these RT gas sensors. We start with an overview of 2D MXenes, explaining their general formula and structures. Then, we present a discussion of their synthesis methods. Following this, the gas-sensing properties of 2D MXenes in pristine, doped, decorated and composite forms are presented. Associated gas-sensing mechanisms are also presented. Finally, we conclude with a summary of current challenges and future prospects. In this review paper, we only present the resistive MXene gas sensors at RT. However, it should be noted that MXenes also can be used for fabrication of other types of gas sensors. Also, there are some composites employing MXenes and working at higher temperatures. Herein, the focus was only on the resistive MXene-based gas sensors which are able to work at RT.

## 2. MXenes: A Brief Introduction

Since the discovery of 2D MXenes with nanosheet (NS) morphology by Gogotsi et al. [34] in 2011, MXenes have attracted much interest because of their features such as 2D morphology, high conductance, tunable bandgap, high mechanical flexibility, and hydrophilicity [35,36]. Their general formula is *M_n_*_+1_*X_n_*T*_x_*, in which *M* represents the transition metal (Mo, Ti, Zr, Cr, etc.), *X* stands for “C” or “N” sites, *n* = 1 to 4, and T*_x_* (*x* is variable) indicates surface termination groups such as -H, -O, -OH, -F [37]. 

In these materials, layers of *M* atoms are arranged in a honeycomb-like 2D lattice intervened by X ions in the octahedral sites between adjacent metal layers. The precursors of MXenes are from MAX phases, with the formula *M_n_*_+1_*AX_n_* (MAX), where “*n*” = 1 to 3, M is a transition metal, and *A* is a group 13–16 element. Figure 1a,b show the elements involved in MAX and the MXenes synthesized from the MAX phase, respectively [38]. Various routes are available for synthesizing MXenes [39,40]. However, they are generally produced by etching the MAX phase, in which the A-layer atoms are selectively etched to generate loosely stacked MX layers [41,42]. Mostly, the HF etchant with varying chemical composition is used to synthesis MXenes by the selective etching of Al atoms from Al-MAX phases. However, to achieve successful conversion of the MAX phase into the MXene, the control of the *HF* concentration, the reaction temperature and time are essential, since the M-Al bond strength depends on the type of *M* elements. It should be noted that intense etching lead to formation of defects, affecting the MXene quality. In addition, HF etching converts the MAX phase into accordion-like multilayered MXenes, in which individual NSs are held together by van der Waals forces and hydrogen bonds. During HF etching, the following reactions cause the removal of “*A*” atoms and generation of MXene.
(1)$$M_{n+1}AX_{n}+3HF=M_{n+1}X_{n}+AF_{3}+1.5H_{2}$$
(2)$$M_{n+1}X_{n}\left(s\right)+2HF=M_{n+1}X_{n}F_{2}+H_{2}\left(g\right)$$

Due to the high toxicity of *HF*, MXenes are also synthesized by using fluoride salt-derived in situ *HF* etching, fluoride-free etching, molten salt etching, and electrochemical-derived etching of MAX phases [43,44,45]. MXenes have also been used as templates for the synthesis of other materials [46]. Owing to their large surface area, controllable interlayer spacing, abundant functional groups, and unique electrical properties, MXenes are considered promising materials for realizing RT gas sensors [47,48]. Among MXenes, the most widely used for gas sensors is Ti_3_C_2_T*_x_* MXene due to the following facts: (i) large specific area and numerous terminal functional groups (-OH, -O, and -F) on Ti_3_C_2_T*_x_* MXene can lead to the strong interface chemical connection with semiconductors and form a Schottky junction. (ii) The high metallic conductivity of Ti_3_C_2_T*_x_* assures rapid carrier migration. (iii) The exposed terminal metal sites on MXenes may result in more active reactivity than that of carbon materials [49]. 

One of the major limitations for the practical application of MXenes is their poor oxidation stability under ambient conditions. In fact, exposure to water, air, heat, and light degrades the 2D MXenes into a composite of metal oxide nanoparticles and amorphous carbon. The oxidation of the MXene NSs is generally initiated at surface defects/edges and then propagates to other parts of the nanosheet [43]. The oxidation and/or the state of MXene oxidation can be investigated by Raman spectroscopy [50]. For sensing applications, generally at RT there is no notable oxidation, as the sensing temperature is low. However, at high temperatures, MXene composition changes due to partial oxidation, which leads to changes in the response to the target gas. Therefore, MXenes are mostly used at low temperature due to their relatively low oxidation resistance. Also, for RT MXene-based gas sensors, the long-term stability is acceptable due to working at low temperatures. In the following sections, we discuss the RT gas-sensing properties of MXene sensors. 

### 2.1. Pristine MXene Gas Sensors

For pristine MXene gas sensors, the sensing mechanism relies on three factors. First, the oxygen molecules can be directly adsorbed on the sensor surface and change the amount of charge carriers on the outer surfaces of MXene, leading to formation of a hole accumulation layer (HAL) or an electron depletion layer (EDL) on p- and n-type MXenes, respectively. Upon exposure to target gases, the amount of charge carriers significantly changes, leading to significant modulation of the sensor resistance. Second, since MXenes are generally synthesized by selective etching, some defects will be present on their surfaces which act as favorable sites for adsorption of gas molecules. Third, the surface functional groups on the sensor surface also are considered as potential sites for adoption of target gas. In addition, the high surface area of MXene NSs provides plenty of adsorption sites for incoming gas molecules.

#### 2.1.1. Delaminated MXene Gas Sensors

Two-dimensional Ti_3_C_2_T*_x_* MXene NSs were fabricated by etching Ti_3_AlC_2_ in a mixture of 9 M HCl and LiF (~1 g) at 35 °C for 24 h, and they were delaminated by sonication. Then, they were deposited on a flexible polyimide (PI) substrate for NH_3_ gas sensing at RT [51]. Polymeric substrates are generally employed to realize flexible/wearable sensors owing to their low cost, flexibility, and stretchability. Flexible PI is widely used as a flexible substrate because of its excellent bendability, high thermal stability, and high chemical stability [52,53]. One fabricated sensor exhibited a response of 21% ([(|R_g_ − R_a_|)/R_a_] × 100) to 100 NH_3_ at RT. The p-type semiconducting behavior of Ti_3_C_2_T*_x_* originated from the presence of H_2_O and oxygen on the surface of MXene, which were added during etching. NH_3_ gas was adsorbed by both defects and functional groups, such as -O and -OH, on Ti_3_C_2_T*_x_* MXene. The bonding of NH_3_ was stronger via hydrogen bonds, leading to the transfer of electrons from NH_3_ to Ti_3_C_2_T*_x_*, which combined with the holes inside the Ti_3_C_2_T*_x_* sensor, leading to an increase in the resistance and generation of the sensing signal [51]. 

High-performance gas sensors should have low electrical noise, owing to their high conductivity, and strong signals owing to their strong and abundant adsorption sites [54]. However, it is difficult to satisfy these conditions. In fact, metal oxide gas sensors only show both a high signal and low noise at high temperatures due to the presence of activation energy. On the other hand, highly conductive channel materials are likely to yield low noise but lack the gas adsorption sites required for a high signal. In this regard, MXenes with both high conductivity and a high amount of adsorption sites are promising candidates. In an interesting study, Ti_3_C_2_T*_x_* MXenes were synthesized by etching using a mixture of lithium fluoride (LiF) and 9 M hydrochloric acid (HCl) while stirring. Then, they were delaminated by sonication. The Ti_3_C_2_T*_x_* sensor exhibited high selectivity to hydrogen-bonded gases over acidic gases and showed an empirical limit of detection (LOD) of 50 ppb. The Ti_3_C_2_T*_x_* sensor displayed significantly high signal-to-noise ratios (SNR) over the entire ppb range, and an SNR of 25.6 was obtained for acetone detection at 50 ppb. Both the metallic-like conductivity and presence of many adsorption sites are responsible for the high SNR of the sensor [54].

Because residual ions on the surface of Ti_3_C_2_ during etching can affect the gas adsorption properties of MXene [55], it is necessary to explore the effect of the solution used for etching the MAX phase. Single-layer Ti_3_C_2_ MXenes were prepared by etching the Ti_3_AlC_2_ phase with a mixed solution of NaF/HCl. The sensor displayed a response of 6% ([(|R_g_ − R_a_|)/R_a_] × 100) to 500 ppm NH_3_ gas at RT. Owing to the use of the NaF/HCl solution, the surface of Ti_3_C_2_ was quite clean, as the Na ions were easily removed. This provided more adsorption sites for the NH_3_ gas molecules. Density functional theory (DFT) calculations confirmed that the adsorption energy of NH_3_ was higher than that of other gases; thus, it showed the strongest interaction with the sensing layer [56].

Because the-O and -OH intrinsic groups on the MXene surface act as favorable adsorption sites, their amounts can be increased to enhance the overall gas-sensing performance. In this context, plasma exposure is a highly promising technique with good control over parameters such as power, exposure duration, and atmosphere. In addition, the plasma exposure does not damage the structure of the MXenes. Exposure is particularly advantageous because the delicate MXene layers will not be damaged. In this context, a large number of oxygen functional groups were grafted onto delaminated Ti_3_C_2_T*_x_* MXenes via in situ plasma exposure. Air was first introduced at a flow rate of 400 sccm in a chamber at a vacuum level of 101 mbar. Next, the sensors were irradiated by plasma. The optimal sensor displayed a response of 13.8% ([(|R_g_ − R_a_|)/R_a_] × 100) to 10 ppm NO_2_ at RT. DFT calculations revealed that the oxygen functional groups were associated with increased NO_2_ adsorption energy, thereby enhancing the gas response [57].

The surface groups of the MXenes are responsible for their hydrophilic nature [58]. The introduction of hydrocarbon groups can reduce the hydrophilicity and enhance the sensing performance [59]. The modification was performed by substituting the -F groups with hydroxyls and subsequent treatment with trimethylacetic anhydride. Pristine Ti_3_C_2_T*_x_* exhibited a water contact angle (WCA) of 36.3°, reflecting its hydrophilic nature. After modification, the WCA increased to 78.5°, revealing the less hydrophilic nature of the treated MXene. In addition, the surface area of the treated sample was ~39 m^2^ g^−1^, which was remarkably higher than that of pristine Ti_3_C_2_T*_x_* (~10.35 m^2^ g^−1^). The pristine and treated MXene sensors showed responses of 3 and 8% ([(|R_g_ − R_a_|)/R_a_] × 100) to 20 ppm ethanol at RT. Moreover, the response of the treated sensor to water vapor was reduced by 71% relative to pristine Ti_3_C_2_T*_x_*, due to the less hydrophilic nature of the treated sensor. The treated Ti_3_C_2_T*_x_* sensor was successfully used for alcohol detection via exhaled breath analysis [59].

Surface treatment is a feasible strategy to enhance the stability of MXene-based sensors at RT. 3-Aminopropyl triethoxysilane (APTES), as a silane coupling reagent, can not only decrease the oxidation of MXenes by the addition of a protective layer but can also add additional reactive groups such as -NH_2_ to MXene, which are promising for the detection of acidic gases. In this regard, delaminated Nb_2_CT*_x_* MXene was dispersed in a water and ethanol mixture of 1:9 ratio to provide enough water for a hydrolysis reaction. Subsequently, three different concentrations of APTES (0.1 mL, 0.2 mL, and 0.3 mL) were added to the above mixture. The hydrophilic -NH_2_ group with an electron-donating nature was useful for NO_2_ adsorption. The responses of 0.1, 0.2, and 0.3 mL APTES-functionalized Nb_2_CT*_x_* MXene sensors to 25 ppm NO_2_ gas were 22.5, 31.52, and 26.8% ([(|R_g_ − R_a_|)/R_a_] × 100), respectively, while that of the pristine sensor was only 12.5%. Hence, the addition of APTES to Nb_2_CT*_x_* MXenes enhanced the sensing response via amine functionalization. However, upon 0.3 mL addition of APTES, due to the increase in the number of Si-O-Si groups, the resistance of MXene increased, which limited the charge transfer and decreased the sensor response [60].

The positive response of MXenes to various gases can be attributed to the interlayer swelling. The resistance of the swollen 2D layers increases because of the limited out-of-plane flow of electrons. Therefore, preintercalation is a good method for optimizing the sensing features of MXenes. In this regard, the swelling of Ti_3_C_2_T*_x_* MXenes treated with NaOH upon gas injection was explored using in situ XRD measurements. Ti_3_C_2_T*_x_* MXene swelled upon ethanol injection, whereas no swelling was observed in a CO_2_ atmosphere. The swelling amount depends on the concentration of NaOH ions, and a 0.3 mM NaOH concentration resulted in the largest degree of swelling and the highest response to ethanol [61]. In another related study, first, V_2_CT*_x_* was synthesized by etching V_2_AlC in a mixture of NaF and HCl. Secondly, the as-prepared V_2_CT*_x_* powder was further exfoliated by intercalation with dimethyl sulfoxide (DMSO). Finally, delaminated V_2_CT*_x_* powder was dispersed into a 10 mL NaOH solution (5 M) for 2 h. The alkalized V_2_CT*_x_* sensor revealed an 80-times higher response to 50 ppm NO_2_ gas than the pristine sensor at RT. Because the alkalized V_2_CT*_x_* sensor exhibited a positive gas response to both oxidizing and reducing gases, it was assumed that the sensing mechanism was related to interlayer swelling. The gas molecules diffuse into the layers of V_2_CT*_x_*, causing swelling of the layers. Furthermore, the high ratio of -O/-F surface terminal groups after alkalization is useful for gas sensing [62]. A similar alkalization process using a 5 M NaOH solution was reported for Ti_3_C_2_T*_x_* with enhanced NH_3_ gas-sensing performance [63]. In another study, V_2_CT*_x_* MXene was deposited on a PI substrate and was able to detect 2 ppm hydrogen at RT. The amounts of oxygen and hydroxide surface groups on MXene were higher than -F groups; therefore, the sensor response was good because of the dominance of these surface groups. In addition, the presence of V ions as transition metals is beneficial for sensing H_2_ because of their catalytic activity towards H_2_ gas [64]. 

In another study related to Mo_2_CT*_x_*, first, 2 g of Mo_2_Ga_2_C powder was immersed in a ~25 wt% HF solution at 55 °C for 8 days. The resultant Mo_2_CT*_x_* powder was dried in a vacuum oven. Then, 1 M TMAOH solution was used to delaminate the multilayer Mo_2_CT*_x_*. The delaminated Mo_2_CT*_x_* suspension was then collected after centrifugation. To study the effect of the Mo_2_CT*_x_* amount in the suspension on the sensing performance to toluene gas, five Mo_2_CT*_x_* suspensions (0.011, 0.033, 0.066, 0.36, and 0.66 mg/mL) were prepared. Also, the sonication times were varied between 0.5 to 8 h. For the sensors fabricated from the samples with the concentrations of Mo_2_CT*_x_* increased up to 0.066 mg/mL, many conducting paths were generated on the surface and the sensing area was increased so a higher response was observed. Accordingly, the sensor prepared with a concentration of 0.066 mg/mL exhibited the highest sensing performance. With a further increase in the concentration, the deposited Mo_2_CT*_x_* sensor became thicker, and conducting paths were generated below the surface; hence, the modulation of the conductance was negligible. The optimal sonication time was determined to be 8 h. The sensing mechanism was attributed to the interaction between the benzene rings of the C_7_H_8_ molecules and MoC_2_T*_x_*. Indeed, the benzene ring has a high activity and interacts more strongly with MoC_2_T*_x_*, resulting in a more significant change in the charge carriers inside the sensing layer. Furthermore, the -CH_3_ group in toluene further improved the activity of the benzene ring in the toluene molecules. In addition, according to DFT calculations, the adsorption energy of toluene on MXene was the highest, reflecting the intense interaction between toluene and the MXene surface, which contributed to the higher response to toluene [65].

The detection of methane at RT is difficult because of the high enthalpy of the C-H bonds and their nonpolar nature [66]. The Ti_2_CT*_x_* MXene sensor was used as a CH_4_ gas sensor at RT under visible light illumination. Under visible-light illumination, the sensing response to CH_4_ gas was significantly improved relative to that under dark conditions. The photocatalytic CH_4_ oxidation activity of Ti_2_CT*_x_* was responsible for its enhanced gas response. Under illumination, the generated electrons and holes generated highly active species (O^−^ or OH• radical ions). In addition, the Ti^4+^ ions accepted the generated electrons and were converted to Ti^3+^ centers with high activity towards CH_4_ and the generation of •CH_3_ active species. Then, •CH_3_ reacted with O^−^ or OH• species to produce final products. The released electrons increased the sensor resistance and contributed to the sensing signals [67].

#### 2.1.2. Multilayered MXene Gas Sensors

Generally, van der Waals attractions between MXene NSs lead to self-stacking and agglomeration, which limits the adsorption sites on the sensor surface. Accordion-like Ti_3_C_2_T*_x_* MXenes were synthesized using an HF etching method for acetone sensing. The response of the sensor to 100 ppm acetone was 100% ([(|R_g_ − R_a_|)/R_a_] × 100) at RT, and it was able to detect 250 ppb acetone with a fast response time (t_res_) of 53 s at RT. The high surface area is due to the accordion-like morphology of Ti_3_C_2_T*_x_* as well as the presence of a large number of hydrogen bonds between the functional groups on the MXene surface and acetone vapor attributed to the sensing enhancement. However, the response of the gas sensor in the presence of 60% or higher RH was significantly decreased [68].

Mo-based MXenes, such as Mo_2_CT*_x_*, offer more conductance and higher reactivity than Ti-based MXenes; however, less attention has been paid to them. Three gas sensors based on multilayered Mo_2_CT*_x_* MXenes on glass, crystalline Si (cSi), and porous Si (pSi) substrates were used for CO_2_ sensing. The sensor deposited on glass Si substrate displayed the best response to CO_2_ gas, with a good response of 2.3% ([(|R_g_ − R_a_|)/R_a_] × 100) at RT to 50 ppm CO_2_ and fast t_res_ and recovery times (t_rec_) of 28 and 40 s, respectively [69]. However, at higher temperatures, the sensor deposited on pSi exhibited an enhanced response to CO_2_ gas. The enhanced gas response was justified by the lack of charge transfer from either the cSi or pSi substrates to MXene at RT. However, at higher temperatures, the charge transfer from these substrates to MXene leads to a decrease in resistance, which ultimately contributes to the sensing response. 

Table 1 presents the RT gas-sensing properties of pristine MXene-based gas sensors. They are mostly used for detection of NH_3_, NO_2_, ethanol, acetone and CO_2_ gases at RT.

It should be noted that the morphology of the sensing layer is mostly useful for enhancement of the response towards all gases and it has less effect on the selectivity. In general, MXene gas sensors in pristine form have good selectivity to NH_3_ gas. However, with modification with other materials, such as composite-making, doping or decoration, either selectivity to NH_3_ can be increased or selectivity towards another gas can be increased. Also, it should be noted that in general, the pristine MXene gas sensors suffer from incomplete recovery. However, with different modifications, the recovery rate can be increased to full recovery.

### 2.2. Composite MXene Gas Sensors

Generally, pristine MXene gas sensors suffer from low sensitivity and slow t_rec_, limiting their applications. Composite fabrication is a solution to overcome these limitations. It is well known that composites enhance gas-sensing responses [70,71,72,73]. This is due to the high surface area of composites resulting from NS morphology and the formation of plenty of heterojunctions. The synthesis of MXene-based composites is reviewed in detail in [74]. In this section, we discuss the sensing performances of composite-based MXenes.

#### 2.2.1. MXene–Metal Oxide Composites

The combination of MXenes with metal oxides is a promising strategy for enhancing the RT-sensing properties of the resultant composite, which generally leads to high-performance gas sensors at RT. Therefore, a SnO_2_/Ti_3_C_2_T*_x_* composite was synthesized hydrothermally. A mixture of MXene powder and stannic chloride pentahydrate (SnCl_4_·5H_2_O) was prepared. Then, it was put into a 50 mL Teflon-lined autoclave and heated at 180 °C for 12 h. The fabricated sensor offered a response of 40% ([(|R_g_ − R_a_|)/R_a_] × 100) to 40 ppm NH_3_ at RT, which was higher than a pristine sensor. In the composite, the 2D MXene provided a matrix with high conductivity, which enabled RT sensing. NH_3_ absorption at the defect sites on the MXene surface, as well as the interaction with functional groups, resulted in an enhanced gas response. Furthermore, the formation of heterojunctions between MXene and SnO_2_ NPs, which acted as resistance modulation sources, contributed to the sensing enhancement [75].

The influence of the amount of MXene in the Ti_3_C_2_T*_x_*-SnO_2_ composite was also investigated. Ti_3_C_2_T_X_ MXene (10–40 wt%)-SnO_2_ composites were prepared using a hydrothermal route for NO_2_ sensing at RT. SnO_2_ and Ti_3_C_2_T_X_ (10, 20, 30, and 40 wt%) were sonicated, and 0.32 g of urea was added into it along with the dropwise addition of 40-µL HCl. The obtained solution was put inside an autoclave and heated at 120 °C for 8 h. All the SnO_2_/MXene composite sensors exhibited superior performance to NO_2_ gas relative to pristine MXene. The composite sensor not only had a higher surface area (~25–30 m^2^/g) than pristine MXene (8 m^2^/g) due to combination of NS morphology of MXene with SnO_2_ NPs, but also a higher conductivity relative to pristine SnO_2_ NPs owing to the presence of MXene. The presence of MXene facilitated charge-carrier transport during gas sensing, resulting in faster t_res_ and t_rec_. In addition, the functional groups of the Ti_3_C_2_T_X_ MXenes are favorable sites for gas adsorption. The sensor with the lowest amount of MXene (10 wt%) showed some agglomeration between the SnO_2_ NPs, while the sensor with 20 wt% Ti_3_C_2_T_X_ showed the maximum performance. The response decreased with a further increase in the MXene amount, which was related to the presence of enormous -OH termination groups, hindering the number of active sites for NO_2_ gas. For the optimal gas sensor, because of the sufficient number of gas adsorption active sites and enough charge-carrier transportation as a result of fewer agglomerated SnO_2_ NPs, a high response was observed. The formation of heterojunctions between MXene and SnO_2_ and subsequent modulation of the barrier height in the presence of NO_2_ gas also contributed to the sensing signal (Figure 2). SnO_2_ and MXene have different work functions; hence, upon intimate contact, electrons from MXene move to SnO_2_ to equate the Fermi levels inside of the contacts. This leads to the formation of potential barriers to the flow of electrons in interfaces in air. Upon subsequent exposure to NO_2_ gas, more electrons are abstracted from the sensing layer, and this increases the height of the potential barrier. Accordingly, the resistance will be significantly increased [76].

In another study, Ti_3_C_2_T*_x_* MXene/SnO NSs were prepared via a hydrothermal route. After magnetic stirring of the Ti_3_C_2_T*_x_* MXene solution for 1 h, SnCl_2_·2H_2_O and urea were added into the above solution, and then 30 μL HCl (36–38%) was added dropwise. Then, it was hydrothermally treated at 120 °C for 24 h in a Teflon autoclave. The responses of the fabricated sensor to 200 ppm NH_3_ was 7.8 at RT. t_res_ and t_rec_ were 1 and 2 min, respectively. Figure 3a shows the sensing mechanism of the sensor. Heterojunctions were formed between the n-type SnO NSs and p-type Ti_3_C_2_T*_x_* in air due to differences between the work functions. Accordingly, potential barriers were formed in interfaces between these two materials in air. Subsequent modulation of the heights of these barriers in an NH3 atmosphere resulted in the resistance change (Figure 3b). In addition, the high surface area originating from the 2D nature of both sensing materials and the existence of surface groups resulted in a high sensor response [77].

SnO_2_ NPs (5–10 nm) were dispersed on the Ti_3_C_2_T*_x_* MXene surface during the hydrothermal synthesis. SnCl_4_·5 H_2_O and different volumes (10, 15, 20 and 25 mL) of the few-layered Ti_3_C_2_T*_x_* colloidal solution were poured to form a homogeneous mixture by means of electrostatic self-assembly; the obtained samples were named as 4.8%, 9.2%, 13.1% and 16.8%Ti_3_C_2_T*_x_*-SnO_2_ composites, respectively. Then, CO(NH_2_)_2_ dissolved in various deionized water (30, 25, 20 and 15 mL) was dripped slowly into the mixture and stirred for 0.5 h. The resulting mixed solution was subsequently transferred into a 100 mL Teflon-lined autoclave and kept at 180 °C for 12 h.

A small amount of TiO_2_ was formed during synthesis. A high response of 25% ([(|R_g_ − R_a_|)/R_a_] × 100) to 100 ppm NH_3_ gas was recorded at RT for the optimal sensor with 13.1% MXene. The SnO_2_ NPs formed a loose porous structure and provided a large surface area and abundant active sites for sensing reactions. In addition, TiO_2_ NPs formed along the MXene interlayers, preventing their restacking. The formation of heterojunctions between Ti_3_C_2_T*_x_* MXene, TiO_2_, and SnO_2_ NPs led to resistance modulation during the injection of NH_3_ gas (Figure 4) [78].

An ultrasonic method was used to prepare NiO/Ti_3_C_2_T*_x_* MXene nanocomposites. NiO and MXene powders were dissolved in deionized water and then subjected to ultrasonic treatment for 5 h. Then, it was centrifuged to separate the precipitate, and the NiO/Ti_3_C_2_T*_x_* MXene composite was obtained. The sensor exhibited a high response of 6.13% ([(|R_g_ − R_a_|)/R_a_] × 100) to 50 ppm NH_3_ at RT, which was ~9 times more than that of the pristine MXene sensor. Well-dispersed NiO particles in the interlayers of the accordion-like MXene not only prevented agglomeration but also increased the surface area (Figure 5a). Hence, it improves the absorption and diffusion of NH_3_ molecules. Furthermore, many functional groups on MXene can easily form strong hydrogen bonds with NH_3_ gas. In addition, the high conductivity of MXenes accelerates charge transfer, thereby enhancing the gas response. The formation of heterojunctions also accounted for the sensing enhancement. Figure 5b shows the modulation of the energy bands of the NiO/Ti_3_C_2_T*_x_* MXene nanocomposite sensor in air and in NH_3_ gas [79].

A MXene/NiO composite was synthesized via an in situ precipitation method. A NiSO_4_·6H_2_O and MXene solution was prepared, and then a NaOH aqueous solution was dropped into the solution. After being stirred for 2 h, the precipitates were collected and washed three times with deionized water and three times with ethanol. Then, the samples were dried at 60 °C for 24 h. Finally, the MXene/NiO composite materials were obtained after being calcined at 350 °C for 2 h under N_2_ atmosphere. The sensor response to 50 ppm HCHO gas was 8.8 at RT. Based on FTIR analysis, numerous hydroxyl and other oxygen-containing functional groups were present on the sensor surface, which are important for NH_3_ gas sensing. Also, based on the in situ FTIR analysis (Figure 6), by increasing exposure time of HCHO gas, the peaks at 1352 and 1413 cm^−1^ were attributed to molecularly adsorbed HCHO on the sensing material. The peaks at 1340 and 1557 cm^−1^ were related to the COO^−^ symmetric stretching and asymmetric stretching vibrations of formate species, respectively. The peak at 1510 cm^−1^ was related to the vibrations of formate species. Hence, formate species are intermediate products of HCHO adsorption [80].

Ti_3_C_2_T*_x_* MXene (30–100 wt%)/In_2_O_3_ composite gas sensors were prepared for NH_3_ sensing studies. Initially, In_2_O_3_ was prepared by dissolving In(NO_3_)_3_·xH_2_O in a mixed solution of ethanol and 25 wt% NH_3_·H_2_O and subsequent hydrothermal synthesis at 100 °C for 24 h. Then, In_2_O_3_/MXene with different amounts of MXene was sonicated, and after vacuum drying at 60 °C for 6 h the composites were prepared. The sensor with 40 wt% MXene exhibited a high response of 100% ([(|R_g_ − R_a_|)/R_a_] × 100) to 20 ppm NH_3_ gas, which was 30 times more than the response of pristine MXene. Moreover, the response to NH_3_ increases with increasing RH in the gas mixture. In a humid environment, the ammonia molecules were “solvated” and the released electrons in a humid environment increased the sensor resistance:NH_3(gas)_ + H_2_O_(ads)_ → NH_4_^+^_(ads)_ + OH^−^_(ads)_.(3)

Then, NH_4_^+^ reacted with O_2_^−^_(ads)_ as follows:2NH_4_^+^_(ads)_ + 3O_2_^−^_(ads)_ → 2NO_(ads)_ + 4H_2_O_(ads)_ + e^−^,(4)
NO_(ads)_ + 1/2O_2(gas)_ → NO_2(ads)_(5)

The generation of gaseous nitric oxides was confirmed by in situ diffuse reflectance infrared Fourier transform (DRIFT) spectroscopy during the sensing process. The generated electrons combine with holes on MXene, resulting in an increase in resistance and an enhanced response [81]. In another study dealing with MXene/SnS_2_ heterojunction sensors [82], similar to [81], the response increased under humid conditions. In situ DRIFT analysis confirmed the production of nitric oxides during the NH_3_ reaction on the sensor surface in a humid environment. In addition, the enhanced response to NH_3_ gas was attributed to the formation of heterojunctions between MXene and SnS_2_. DFT calculations demonstrated strong NH_3_ adsorption on the sensor surface.

In_2_O_3_ nanocubes/Ti_3_C_2_T*_x_* MXene composites were prepared using a hydrothermal method. In_2_O_3_ nanocubes were dissolved in ethanol via ultrasonication, and then APTES was poured dropwise into the above solution to make the surface positively charged. Next, Ti_3_C_2_T*_x_* MXene solution was slowly added to the above solution, and then it was put into a Teflon-lined autoclave. After the hydrothermal reaction at 120 °C for 14 h, the In_2_O_3_/Ti_3_C_2_T*_x_* MXene composites were obtained. The In_2_O_3_ nanocubes were well dispersed between the Ti_3_C_2_T*_x_* MXene NSs layers (Figure 7a). The sensor exhibited a high response of 29.6% ([(|R_g_ − R_a_|)/R_a_] × 100) to 5 ppm methanol gas at RT. In addition, t_res_ and t_rec_ were very fast (6.5/3.5 s). The sensing materials had a mesoporous structure; thus, the target gas easily diffused into the deeper parts of the sensor. The existence of many functional groups on MXene which acted as active sites for the adsorption of methanol, as well as the generation of Schottky junctions between Ti_3_C_2_T*_x_* MXene and the In_2_O_3_ nanocubes, accounted for the sensing improvement (Figure 7b) [83]. Due to the difference between the work functions of Ti_3_C_2_T*_x_* and In_2_O_3_ materials, electrons move from Ti_3_C_2_T*_x_* to In_2_O_3_, resulting in band bending and formation of potential barriers in interfaces. In a methanol atmosphere, the height of these barriers changes due to the return of electrons to a sensing layer, leading to modulation of the electrical resistance.

In other interesting research, the MOF-derived hollow In_2_O_3_ microbutes (2–5 µm) were attached on the exfoliated Ti_3_C_2_T*_x_* MXene. Initially, Indium nitrate hydrate and 1,4-benzenedicarboxylic acid (H_2_BDC) were dissolved in N,N-dimethylformamide (DMF) under mechanical stirring to form a homogeneous solution at RT. Next, Ti_3_C_2_T*_x_* MXene solution (0.1 g/mL) was added to the above solution. Subsequently, it was transferred into the oil bath and the temperature was maintained at 120 °C for 30 min. To obtain the final MOF-derived In_2_O_3_/Ti_3_C_2_T*_x_* MXene composite material, it was annealed at 500 °C for 3 h to remove the organic template. The sensor displayed a response of 60.6% ([(|R_g_ − R_a_|)/R_a_] × 100) to 5 ppm NH_3_ gas at RT, along with a fast t_res_/t_rec_ of 3/2 s and outstanding selectivity. Figure 8a shows the possible sensing mechanism for NH_3_ gas. The work function of MIL-In_2_O_3_ is 4.28 eV and that of Ti_3_C_2_T*_x_* is 4.79 eV. Hence, upon contact, electrons move from MIL-In_2_O_3_ to Ti_3_C_2_T*_x_* to equate the Fermi levels. This results in band bending and the formation of potential barriers in interfaces. In a NH_3_ atmosphere, the electrons come back to the surface of the sensor, resulting in changes of potential barriers and changes in the flow of electrons. Accordingly, the resistance is modulated, which contributes to the sensing signal (Figure 8b,c). Hence, the formation of p-n heterojunctions, good intrinsic sensing properties of In_2_O_3_, high surface area provided by MXene, presence of numerous surface groups on MXene, and high mobility of carriers in MXene all contributed to the sensing enhancement. Based on the DFT calculations, the adsorption energies of NH_3_ molecules on the optimized configurations of In_2_O_3_ and Ti_3_C_2_T*_x_* MXene (Figure 8d,e) were −8.1 and −3.7 eV, respectively, reflecting the fact that In_2_O_3_ was the main source of the resistance change in the gas sensor [84].

α-Fe_2_O_3_/Ti_3_C_2_T*_x_* MXene composite with a high surface area of 47.8 m^2^/g was hydrothermally synthesized at 120 °C/14 h using α-Fe_2_O_3_ nanocubes and Ti_3_C_2_T*_x_* MXene. The response to 5 ppm acetone was 16.6% ([(|R_g_ − R_a_|)/R_a_] × 100), along with very fast t_res_ and t_rec_ (5/5 s) at RT, which were better than those of pristine sensors. Apart from the formation of heterojunctions between Fe_2_O_3_ and MXene, the flake-like morphology decreased the agglomeration of α-Fe_2_O_3_ and promoted the adsorption of gas molecules due to the higher surface area. In addition, the presence of defects and functional groups on Ti_3_C_2_T*_x_* MXene facilitated the interaction between acetone and the gas sensor. Based on DFT calculations, the band gaps of Fe_2_O_3_ and MXene were determined to be 2.2 and 0 eV, respectively, reflecting the high metallic conductivity of MXene. Hence, Ti_3_C_2_T*_x_* MXene with high conductivity was able to easily accept electrons, resulting in a significant resistance change of the α-Fe_2_O_3_/Ti_3_C_2_T*_x_* MXene composite. Furthermore, the adsorption energy of acetone on α-Fe_2_O_3_/Ti_3_C_2_T*_x_* MXene heterojunction was −6.7 eV, which was higher than that on pristine α-Fe_2_O_3_ (−0.2 eV) and Ti_3_C_2_T*_x_* (−5.9 eV) sensors [85].

A rose-like α-Fe_2_O_3_/Ti_3_C_2_T*_x_* MXene composite was prepared using hydrothermal synthesis at 120 °C/14 h for NH_3_ sensing. The surface areas of composite, pristine α-Fe_2_O_3_, and Ti_3_C_2_T*_x_* MXene were 29.2, 9.3, and 1.4 m^2^/g, respectively. The enhanced surface area was related to the special morphology of the composite. The composite sensor offered a response of 18.3% ([(|R_g_ − R_a_|)/R_a_] × 100) and fast t_res_ and t_rec_ of <2.5 s to 5 ppm NH_3_ gas at RT. The good response of the sensor was attributed to high surface area, the presence of abundant functional groups, as well as the formation of α-Fe_2_O_3_/Ti_3_C_2_T*_x_* MXene heterojunctions in air and subsequent modulation in a NH_3_ gas atmosphere (Figure 9). The work function of α-Fe_2_O_3_ is 5.9 eV and that of Ti_3_C_2_T*_x_* is 4.79 eV. Accordingly, when they are in contact, electrons from Ti_3_C_2_T*_x_* flow to α-Fe_2_O_3_, while the holes move in different directions to equate the Fermi levels. This results in the formation of heterojunctions with band bending and the generation of potential barriers. Accordingly, the flow of electrons becomes difficult in air. In a NH_3_ atmosphere and upon the release of electrons, the heights of these potential barriers change and this eventually leads to a change of the resistance. In addition, based on the Knudsen diffusion theory, NH_3_ gas, with a lighter mass relative to the other tested gases, showed a fast diffusion rate, which resulted in the selectivity of the sensor to this gas [86].

However, MXene-based sensors that can detect ppb levels in gases have rarely been reported. In this context, a Ti_3_C_2_T*_x_*/WO_3_ nanocomposite with a large surface area of 36 m^2^/g was synthesized hydrothermally at 180 °C/24 h. It displayed a high response of 78% ([(|R_g_ − R_a_|)/R_a_] × 100) to 200 ppb NO_2_ at RT, which was higher than the bare WO_3_ sensor (9.8%) with short t_res_ (3 min) and t_rec_ (75 s). In addition to the modulation of the resistance in the interface areas between MXene and WO_3_ due to the formation of heterojunctions, the high surface area and porous structure of the composite accounted for the sensing enhancement. Furthermore, the highly conductive nature of the MXene phase facilitates charge flow and diffusion of gas inside the sensor [87].

WO_3_ NPs attached to Ti_3_C_2_T*_x_* NSs were synthesized using ultrasonication for 3 h at a frequency of 22 kHz and power of 150 W. The sensor with 50 wt% WO_3_ exhibited a high response of 22.3% ([(|R_g_ − R_a_|)/R_a_] × 100) to 1 ppm NH_3_ at RT, which was more than 15 times higher than the pristine Ti_3_C_2_T*_x_* sensor. The porous structure with a specific surface area of 6.129 m^2^/g was beneficial for the diffusion of gas molecules and enhanced response. In addition, the defects in WO_3_ provide additional sites for the adsorption of NH_3_ molecules on the composite surface. In addition, the functional groups on MXene act as adsorption sites for NH_3_ gas molecules. Finally, the formation of p-n heterojunctions is attributed to the sensing mechanism [88].

A Ti_3_C_2_T*_x_* MXene/CuO composite was synthesized using a solvothermal method at 120 °C/14 h for RT NH_3_ sensing. The sensor exhibited a high response of 46.7% ([(|R_g_ − R_a_|)/R_a_] × 100) to 5 ppm NH_3_ with fast t_res_ (12 s) and t_rec_ (25 s). The surface area of the composite was 30.94 m^2^/g, which was higher than that of pristine CuO (~25.55 m^2^/g) and Ti_3_C_2_T*_x_* MXene (~2.75 m^2^/g). This resulted in more adsorption sites for incoming NH_3_ gas molecules. In addition, the formation of p-p heterojunctions between the two components and the presence of surface functional groups on MXene, which formed strong hydrogen bonds with NH_3_ gas, contributed to resistance modulation. Based on density functional theory (DFT) calculations, the adsorption energy of NH_3_ on CuO was larger than that on Ti_3_C_2_T*_x_* MXene, confirming that gas sensing was mostly governed by CuO [89]. Ti_3_C_2_T*_x_*/CuO nanocomposites with a mesoporous nature and high surface area (26.9 m^2^/g) were prepared using a hydrothermal approach at 150 °C/12 h. The sensor exhibited a response of 57% ([(|R_g_ − R_a_|)/R_a_] × 100) to 50 ppm NO_2_ at RT. The enhanced gas-sensing performance is related to the formation of CuO-Ti_3_C_2_T*_x_* heterojunctions and the presence of more oxygen vacancies relative to pristine Ti_3_C_2_T*_x_* [90].

Accordion-like V_2_CT*_x_* MXene was synthesized by the selective etching of V_2_AlC, and V_2_CT*_x_*/V_3_O_7_ nanocomposites were prepared by the partial conversion of V_2_CT*_x_* to V_3_O_7_ due to oxidation at different temperatures (50–250 °C). The V_2_CT*_x_*/V_3_O_7_ nanocomposite obtained at 250 °C was used for sensing studies. The sensor exhibited a response of 16% ([(|R_g_ − R_a_|)/R_a_] × 100) to 100 ppm NO_2_ at RT. The partial oxidation of V_2_CT*_x_* led not only to an increase in the response of the V_2_CT*_x_*/V_3_O_7_ material but also to some change in its selectivity. In fact, the pristine sensor exhibited the highest response to NH_3_ gas, while the composite showed the highest response to NO_2_ gas at RT. The increase in the response of the nanocomposite sensor was related to the formation of V_2_CT*_x_*/V_3_O_7_ heterojunctions, the presence of defects in MXene, and the high surface area after oxidation [91].

Urchin-like V_2_CT*_x_*/V_2_O_5_ MXene were produced by hydrothermal synthesis of V_2_CT*_x_* MXenes at 90 °C/5 days, followed by transformation into urchin-like V_2_CT*_x_*/V_2_O_5_ by subsequent annealing at 450 °C. The sensor exhibited an almost three-times higher response to 15 ppm acetone than pristine V_2_CT*_x_* MXene at RT. The presence of H-bonds in V_2_CT*_x_* MXene, a high surface area with an urchin-like morphology, synergistic effects between the V_2_C and V_2_O_5_ MXene sensors, and high charge-carrier transport in MXene accounted for the enhanced gas response [92].

In contrast to previous studies on oxidized MXenes, in which oxidation was carried out on assembled films, MXene flakes were oxidized in an aqueous solution to decorate them with oxides and to form maximal Schottky barriers after the process. The sensor, oxidized for 8 h, displayed a response of 175% ([(|R_g_ − R_a_|)/R_a_] × 100) to 5 ppm NO_2_ gas at RT. The formation of interflake Schottky barriers in the TiO_2_/Ti_3_C_2_ system has been reported as the main reason for the improved gas response [93]. The oxidation of the Ti_2_CT*_x_* MXenes was investigated at different temperatures (20–447 °C). At 316 °C, oxidation of Ti_2_CT*_x_* MXene started, and at 447 °C it was completely oxidized to TiO_2_. The sample oxidized at 316 °C showed better sensing performance relative to the pristine sample. It exhibited the highest response of 61% ([(|R_g_ − R_a_|)/R_a_] × 100) to 100 ppm NH_3_ gas at RT. However, their selectivity for NH_3_ gas is poor. The increase in the sensing response was attributed to the formation of p-n heterojunctions at the Ti_2_CT*_x_*/TiO_2_ interface. At higher temperatures (T > 350 °C), MXene was not stable. Therefore, MXene sensors are better suited for use at low temperatures [94].

Ti_3_C_2_T*_x_* MXene/ZnO nanorod (NR) composite synthesized by a chemical route with a surface area of 146.8 m^2^/g showed a very high response of 346% ([(|R_g_ − R_a_|)/R_a_] × 100) to 0.2 ppm NO_2_ at RT under UV light, and its t_res_ and t_rec_ were 17 and 24 s to 50 ppb NO_2_, respectively. Both MXene and ZnO were photoexcited under UV illumination. The photogenerated electrons in ZnO led to an expansion of the conduction paths inside the ZnO NRs. Moreover, metallic Ti_3_C_2_T*_x_* MXene served as an electron collector with the formation of Ohmic junctions and further contributed to the generation and separation of photocarriers in the ZnO NRs. Hence, more free carriers were available for gas-sensing reactions under UV irradiation. These photogenerated electrons were abstracted by NO_2_ gas, resulting in more remarkable variations in the conduction path width compared with the case of the hybrid. Furthermore, the mesoporous structure of the sensor provides many adsorption sites and gas-diffusion channels for NO_2_ gas [95].

A Ti_3_C_2_T*_x_*-ZnO NS composite was fabricated using a simple sonication approach for 30 min at a power of 100 W. The sensor displayed a high response of 367.63% ([(|R_g_ − R_a_|)/R_a_] × 100) to 20 ppm NO_2_ at RT under UV illumination. The large surface area of the gas sensor (9.70 m^2^/g) resulting from its 2D morphology and surface groups of Ti_3_C_2_T*_x_* along ZnO oxygen vacancies provided numerous adsorption sites for NO_2_ gas. Furthermore, the formation of Schottky junctions between Ti_3_C_2_T*_x_* and ZnO NSs and the photogenerated charge carriers of ZnO under UV light resulted in interactions between NO_2_ and ZnO NSs. Based on DFT calculations, the main adsorption sites for NO_2_ were on the ZnO NSs, and Ti_3_C_2_T*_x_* acted as a conductive path to accelerate the flow of charges, resulting in the fast dynamics of the gas sensor [96].

In situ growth of (001) TiO_2_ onto 2D Ti_3_C_2_T*_x_* MXene was performed using hydrothermal synthesis at 160 °C/2–16 h for sensing studies. Under UV light, the sensor prepared from the composites treated for 2 h showed ~34 times more response to 30 ppm NH_3_ than that of pristine Ti_3_C_2_T*_x_*. UV light excited electron-hole pairs on the surfaces of (001) TiO_2_ and Ti_3_C_2_T*_x_*; hence, more electrons and holes were available for sensing in the reactions. TiO_2_ with a highly active (001) crystal plane provided efficient photogeneration under UV light, while Ti_3_C_2_T*_x_* stored holes through the Schottky junction with TiO_2_, which increased the separation of electron-hole pairs, thereby improving the NH_3_ sensing performance. In addition, an integrated circuit alarm system was designed to successfully detect the decay process of fresh fish (Figure 10a–d) [97].

The ZnSnO_3_/Ti_3_C_2_T*_x_* MXene composite hydrothermally synthesized at 150 °C for 24 h exhibited a high response of 194.7% ([(|R_g_ − R_a_|)/R_a_] × 100) towards formaldehyde gas at RT with fast t_res_ and t_rec_ of 6.2 s and 5.1 s, respectively. The surface area of pristine ZnSnO_3_ was 20.62 m^2^/g, and it was increased to 28.39 m^2^/g for the composite sample, where ZnSnO_3_ nanocubes were uniformly dispersed on Ti_3_C_2_T*_x_* MXene NSs. In addition, the functional groups on the Ti_3_C_2_T*_x_* MXene NSs provided numerous active sites for the adsorption of formaldehyde molecules. The formation of heterojunctions at the interface between ZnSnO_3_ and Ti_3_C_2_T*_x_* MXene was attributed to the modulation of the resistance. In addition, the fast dynamics are related to the high charge mobility of MXene [98].

#### 2.2.2. MXene-TMD Composites

Two-dimensional transition metal dichalcogenides (TMDs) have high surface areas, abundant adsorption sites, and high surface reactivities; therefore, their composites with MXenes are promising for sensing studies [99,100]. In a relevant study, a Ti_3_C_2_T*_x_*-WSe_2_ composite was chemically prepared and the fabricated sensor displayed a response of 9% ([(|R_g_ − R_a_|)/R_a_] × 100) to 40 ppm ethanol at RT. In addition, fast t_res_ (9.7 s) and t_rec_ (6.6 s) were recorded. The enhanced response to ethanol gas is related to the numerous heterojunctions generated between Ti_3_C_2_T*_x_* and WSe_2_. The enhanced response to ethanol gas is related to the numerous heterojunctions generated between Ti_3_C_2_T*_x_* and WSe_2_. In heterojunctions, band bending occurs, and as a result, potential barriers will be formed between two materials, leading to difficulty of flow of the charge carriers. Upon injection of the target gas, the height of potential barriers changes, contributing to significant resistance changes in heterojunctions. More heterojunctions result in higher modulation of the sensor resistance. In addition, after 1000 bending cycles, the performance not only did not decrease, but also slightly increased owing to the creation of microcracks and wrinkles by the strain forces, which acted as adsorption sites [101].

The influence of the electrode type on sensing performance was explored. A flexible paper-based sensor using a Ti_3_C_2_T*_x_*/WS_2_ composite was fabricated using either a Ti_3_C_2_T*_x_*-MXene electrode (ME) or a Au electrode (AE). The ME + Ti_3_C_2_T*_x_*/WS_2_ gas sensor exhibited the highest response of 15.2% ([(|R_g_ − R_a_|)/R_a_] × 100) to 1 ppm NO_2_ gas at RT, owing to the formation of Ohmic contact between the sensing layer and ME, in contrast to the Schottky contact formed between the sensor and AE. When Au and Ti_3_C_2_T*_x_*/WS_2_ were in contact, the formation of Schottky potential barriers prevented the transport of charges between the two materials, and only a small number of carriers were able to cross the junction. In contrast, when Ti_3_C_2_T*_x_*/WS_2_ contacted the ME, the height of the barrier between the nonmetal ME and the Ti_3_C_2_T*_x_*/WS_2_ sensor was much lower, which allowed the easy transport of charge carriers across the junction. Furthermore, the flexible 2D ME has a large specific surface area and offers adequate adsorption and reaction sites for oxygen and the target gas. In addition, numerous surface groups are present on the ME surface, which affect the sensing performance. Finally, the excellent conductivity of Ti_3_C_2_T*_x_* accelerated the electron flow during the sensing process and shortened t_rec_. The optimized sensor showed good flexibility by maintaining its performance even after bending 500 times by 60° [102]. A MoS_2_/Ti_3_C_2_T*_x_* composite was prepared using a hydrothermal method [103]. The SEM micrographs of the components and composite are shown in Figure 11a–d. Ti_3_AlC_2_ exhibits a laminated structure. After etching in HF, the morphology of Ti_3_C_2_T*_x_* MXene comprised an accordion-like structure (Figure 11b). The MoS_2_ sample showed a nanoflower morphology (Figure 11c). Moreover, Figure 11d shows that the MoS_2_ nanoflowers were on and between the Ti_3_C_2_T*_x_* interlayers, forming a 3D interconnected network structure with high surface areas, which is beneficial for gas-sensing applications.

The sensor exhibited a response of 45% ([(|R_g_ − R_a_|)/R_a_] × 100) to 20 ppm NH_3_ gas at RT. The specific surface area of MoS_2_/Ti_3_C_2_T*_x_* (23.5 m^2^/g) was enhanced relative to pristine counterparts due to the small flake size and template-conformable MoS_2_ NSs, which enhanced the amount of active sites and increased the flow of electrons. The formation of MoS_2_-Ti_3_C_2_T*_x_* MXene heterojunctions and the presence of a large number of S- and Mo-terminated edges enhanced the interaction of NO_2_ gas with the sensing material. Based on DFT calculations, the adsorption of NO_2_ was more favorable than that of other gases owing to its large adsorption energy, leading to an enhanced response of the sensing device to NO_2_ gas [103].

The Etched MoS_2_ NSs were grown on Ti_3_C_2_T*_x_* MXene via hydrothermal synthesis at 210 °C for 18 h (Figure 12). The composite sensor exhibited a high response of 65.6% ([(|R_g_ − R_a_|)/R_a_] × 100) to 100 ppm NO_2_ gas at RT, which is higher than that of MoS_2_. The composite with a 3D uniform morphology had a large number of active sites. Furthermore, the sensing response is attributed to the presence of fast channels for carrier flow between MoS_2_ and Ti_3_C_2_T*_x_* MXene [104].

T_3_C_2_T*_x_*/WS_2_ nanocomposites with different amounts of WS_2_ were prepared using a simple chemical route for NO_2_ sensing. The response to NO_2_ depended on the amount of WS_2_ in the composite, and the sensor with 50% WS_2_ in the composite revealed the best response of 55.6% ([(|R_g_ − R_a_|)/R_a_] × 100) to 2 ppm NO_2_ gas at RT under visible light, which was ~3.2 times higher than that under the dark conditions (17.4%). The metal-like conductivity of MXene improves the separation and transfer of photogenerated electron-hole pairs. In addition, the large contact areas of the 2D/2D heterostructures improved the carrier transportation under light illumination. However, the response decreases gradually, because the photogenerated carriers are captured by the defects inside and at the interfaces between the two materials, leading to the recombination of electrons and holes and a decrease in the sensing performance [105].

Both Ti_3_C_2_T*_x_*-WO_3_ (10, 20, and 30 wt%) and Ti_3_C_2_T*_x_*-MoS_2_ (10, 20, and 30 wt%) composites were fabricated using hydrothermal synthesis at 170 °C for 8 h, for NO_2_ gas sensing studies at RT. The initial Ti_3_AlC_2_ MAX phase exhibited densely layered structures (Figure 13a). After HF etching, the Ti_3_C_2_ MXene exhibited an accordion-like morphology (Figure 13b). Figure 13c shows the morphology of MoS_2_. Figure 13d shows the morphology of the Ti_3_C_2_-MoS_2_ composite, in which MoS_2_ is located on both the surface and the interlayers of Ti_3_C_2_, reflecting intimate contacts between them. The Ti_3_C_2_-MoS_2_ composite sensor showed a response of 35.8% ([(|R_g_ − R_a_|)/R_a_] × 100) to 10 ppm NO_2_ gas at RT, and overall, the responses of Ti_3_C_2_-MoS_2_ composites to NO_2_ gas were higher than those of Ti_3_C_2_-WO_3_ composites. However, the high amount of MoS_2_ (30 wt%) in the composite limited the number of adsorption sites on MXene, causing a decrease in the gas response. The formation of heterojunctions between MoS_2_ and MXene, along with the presence of numerous surface groups on MXene, contributed to the sensor response [106].

#### 2.2.3. MXene-Conducting Polymers Composites

Conducting polymers (CPs) are promising materials for gas sensors because of their high conductivity, possibility of working at RT, tunable chemical composition, easy doping, and low price [107,108,109]; therefore, they can be used with MXene to boost the RT gas-sensing properties of the resultant composite. A sensor was fabricated for RT ammonia sensing by the in situ polymerization of PEDOT and PSS on Ti_3_C_2_T*_x_* MXene. The sensor showed a high response of 36.6% ([(|R_g_ − R_a_|)/R_a_] × 100) to 100 ppm of NH_3_ with t_res_ and t_rec_ of 2 min and 40 s, respectively. In addition, the sensor on the flexible PI substrate exhibited good mechanical flexibility by maintaining its performance at different bending angles. Charge flow occurred between the NH_3_ molecules and the sensor surface, leading to a change in the electrical conductivity. Furthermore, the high specific surface area of the composite, along with π = π interactions, increased the concentration of charge carriers [110].

A Ti_3_C_2_T*_x_* MXene/urchin-like polyaniline (PANI) composite was produced using a template method by employing sulfonated PS nanosphere templates and in situ polymerization on flexible polyethylene terephthalate (PET). The sensor disclosed a high response of 3.70 to 10 ppm NH_3_ at RT, which was higher than that of the pristine sensor. The enhanced sensing was related to the hollow urchin-like morphology of PANI and the NS morphology of Ti_3_C_2_T*_x_*, both of which were beneficial for providing more adsorption sites for NH_3_ gas. Second, Schottky heterojunctions were generated by the intimate contact between PANI and the Ti_3_C_2_T*_x_* NS, which shortened the diffusion length for charges and led to fast charge flow. Furthermore, the degree of protonation of PANI increased through its connection with the Ti_3_C_2_T*_x_* NS. The increased -NH_2_^+^ and = NH^+^ groups in the composite led to an enhanced response to NH_3_ gas. As NH_3_ is an indicator of meat freshness, the fabricated sensor was successfully used to evaluate pork meat freshness. After 36 h, the sensor was able to indicate an increase in NH_3_ concentration in the meat, confirming spoilage [111]. PANI NPs were decorated with Ti_3_C_2_T*_x_* NSs via in situ polymerization. The sensor displayed a response (ΔI/I_0_) of 40 to 200 ppm ethanol gas at RT. In addition, it exhibited good mechanical flexibility; under bending from 0° to 120°, it exhibited almost the same performance, demonstrating good flexibility. In particular, under bending to ~120° it showed a high response of 27.4% ([(|R_g_ − R_a_|)/R_a_] × 100) to 150 ppm ethanol. In addition, the t_res_ and t_rec_ were 0.6 and 0.8 s, respectively, after bending. Based on DFT calculations, the adsorption energies of −0.985, −0.689, and −0.544 were calculated for OH-terminated Ti_3_C_2_, O-terminated Ti_3_C_2_, and F-terminated Ti_3_C_2_, respectively. This demonstrates that the OH-terminated Ti_3_C_2_ had the strongest binding energy for ethanol [112].

A Ti_3_C_2_T*_x_* composite with conjugated polymers (poly[3,6-diamino-10-methylacridinium chloride-*co*-3,6-diaminoacridine-squaraine], PDS-Cl) and polarly charged nitrogen was used for H_2_S gas sensing. The sensor with 10 wt% MXene exhibited a response of 2% ([(|R_g_ − R_a_|)/R_a_] × 100) to 5 ppm H_2_S at RT. Charge transfer-induced modifications in carrier density are responsible for H_2_S gas sensing. By opening the interlayer spaces of MXene by the polymer, the number of accessible active sites increased and intercalation became easier, leading to an enhancement of the charge-transfer effect. Furthermore, the MXene surfaces contain numerous surface groups that have a remarkable effect on gas adsorption [113].

Cationic polyacrylamide (CPAM) with abundant surface groups can form hydrogen bonds with NH_3_, enhancing its response to this gas. The CPAM/Ti_3_C_2_T*_x_* MXene composite was prepared for gas sensing at RT. The sensor displayed a response of 25% ([(|R_g_ − R_a_|)/R_a_] × 100) to 100 ppm NH_3_ gas at RT. In addition, under bending conditions, the sensor performance did not change significantly, and under the bent state (60°) for 140 cycles, the sensor still showed no degradation in performance. With an increase in the bending angle, the current of the CPAM/Ti_3_C_2_T*_x_* sensor was slightly reduced because of the presence of CPAM, which allowed the Ti_3_C_2_T*_x_* NSs to bond together and show good flexibility [114].

#### 2.2.4. Ternary Composites

MXene-based ternary composites have been studied less for gas-sensing applications than binary composites because of the complexity of the synthesis procedure and the need for optimization of the three components. However, they exhibit superior sensing properties because there are more resistance-modulation sources inside the sensing materials.

A hamburger-like SnO-SnO_2_/Ti_3_C_2_T*_x_* MXene nanocomposite was hydrothermally prepared at 120 °C for 8 h. It revealed a high response of 12.1 to 100 ppm acetone at RT, which was higher than that of the pristine sensors. Moreover, it revealed a t_rec_ of 9 s. The improved response was related to the higher surface area of the SnO-SnO_2_-Ti_3_C_2_T*_x_* MXene composite (46.7 m^2^/g), relative to pristine Ti_3_C_2_T*_x_*, (13.6 m^2^/g), and SnO-SnO_2_ (38.6 m^2^/g) gas sensors. Furthermore, there are more resistance modulation sources in the nanocomposite than in the other sensors. Figure 14a shows the energy levels of the SnO-SnO_2_/Ti_3_C_2_T*_x_* sensor components before contact, showing different work functions of these materials. Upon intimate contact, the electrons move from MXene and SnO_2_ to SnO to equate the Fermi levels (Figure 14b,c). This leads to the formation of a hole accumulation layer (HAL) and an electron depletion layer (EDL) on the MXene and SnO_2_ sides, respectively [115]. In an acetone atmosphere (Figure 3d), narrowing of both the HAL and EDL led to the modulation of the sensor resistance, contributing to the sensing feedback.

A Ti_3_C_2_T*_x_* MXene composited with a marigold flower-like V_2_O_5_/CuWO_4_ heterojunction was hydrothermally synthesized at 180 °C/12 h for NH_3_ gas sensing (Figure 15). It showed a high response of 53.5 to 51 ppm NH_3_ at RT with t_res_ and t_rec_ of 100 and 115 s, respectively. After contact of the Ti_3_C_2_T*_x_* MXene with V_2_O_5_/CuWO_4_ in air, a Schottky barrier was formed (Figure 15). Therefore, electrons flowed from n-type V_2_O_5_/CuWO_4_ to MXene. In addition, MXene acted as a pathway for electrons to move through Ti_3_C_2_T*_x_*/V_2_O_5_/CuWO_4_ more easily than through V_2_O_5_/CuWO_4_, and the modulation of the barrier height in the NH_3_ gas contributed to the sensing mechanism. Moreover, Ti_3_C_2_T*_x_* MXene facilitates electron flow to the electrodes [116].

Ternary 2D Ti_3_C_2_T*_x_* MXene@TiO_2_/MoS_2_ composites were prepared using the hydrothermal method for NH_3_ sensing at RT. It showed a response of 164% ([(|R_g_ − R_a_|)/R_a_] × 100) to 100 ppm NH_3_ gas at RT, which was higher than that of the pristine sensor counterparts. The improved sensing of NH_3_ was attributed to the layered nanostructure with a unique morphology and p-n heterojunctions. Furthermore, DFT studies indicated that NH_3_ was able to transfer more charge to the composite surface than to pristine Ti_3_C_2_T*_x_* MXene and MoS_2_, resulting in a higher modulation of the resistance [117].

A 2D Ti_3_C_2_T*_x_* MXene-MoO_2_/MoO_3_ NSs composite was fabricated using the hydrothermal method at 180 °C/10 h for ethanol detection at RT. It revealed a high response of 19.77 to 200 ppm to ethanol, and fast t_res_ and t_rec_ (46 s/276 s). The high surface area (13.54 m^2^/g) and abundant surface groups on MXene provided more active sites for the adsorption of oxygen and ethanol molecules. Furthermore, as revealed in Figure 16a–d, the generated p-n and n-n heterojunctions remarkably improved the carrier migration rate and shortened t_res_ and t_rec_. MXene, MoO_2_ and MoO_3_ have different work functions of 3.9, 6.5 and 2.9 eV, respectively. When they are contacted, the electrons move from MXene and MoO_3_ to MoO_2_ to equate the Fermi levels inside of contacts. This results in formation of Schottky barriers in interfaces between MXene and MoO_2_ and n-n heterojunctions at interfaces between MoO_2_ and MoO_3_ in air. When the sensor is exposed to ethanol, the electrons are released back to the sensor surface and the height of Schottky barriers and heterojunctions decreases, which finally lead to the resistance modulation of the sensor. Finally, the electrical conductivity of MXene decreases the resistance and sensing temperature [118].

Various nanocomposites such as MXenes with GO, ZnO, CuO, GO/ZnO, GO/CuO, ZnO/CuO, and GO/ZnO/CuO have been hydrothermally synthesized for NH_3_ sensing at RT. Among them, Ti_3_C_2_T*_x_* MXene/GO/CuO/ZnO with an optimal ratio of 9:1:5:5 exhibited the best NH_3_ gas sensing without resistance drift. The response to 200 ppm was 96% ([(|R_g_ − R_a_|)/R_a_] × 100) along with good humidity independence. The improved sensing response was related to the generation of multiple p-n and p-p heterojunctions, as well as the presence of many functional groups on the surfaces of MXene and GO [119].

A 3D Ti_3_C_2_T*_x_* MXene/rGO/SnO_2_ aerogel was fabricated using a facile solvothermal approach at 140 °C for 24 h. It exhibited a response of 54.97% ([(|R_g_ − R_a_|)/R_a_] × 100) to 10 ppm formaldehyde at RT. In addition, it indicated short t_res_ and t_rec_ (2.9 and 2.2 s) along with high stability. The high surface area of 103 m^2^/g and the generation of p-n junctions between rGO and SnO_2_ and p-p junctions between MXene and rGO contributed to the sensing mechanism. Based on DFT calculations, the adsorption energy of HCHO on Ti_3_C_2_T*_x_* MXene/rGO/SnO_2_ was −5.7 eV, which was larger than that for other sensors [120].

#### 2.2.5. Other MXene-Based Composites

A hollow nanofiber GaN/Ti_3_C_2_T*_x_* composite was synthesized by hydrothermal nitridation at 120 °C/12 h. Ti_3_C_2_T*_x_*, which has metallic properties, acts as a conductive channel and decreases the overall resistance at RT. In addition, Ti_3_C_2_T*_x_* accelerated charge flow during the sensing reactions, resulting in fast sensor dynamics at RT. Accordingly, the response of the composite sensor to 50 ppm NH_3_ was 3.5 times higher than that of the bare Ti_3_C_2_T*_x_*. The large specific surface area and unique hollow porous morphology of the GaN NFs provide sufficient adsorption sites for NH_3_ gas. The formation of p-n Ti_3_C_2_T*_x_*-GaN heterojunctions is beneficial for resistance modulation. The responses to NH_3_ were not affected by 20–80%RH. At high humidity, the sensor was covered with multilayered physisorbed water, leading to the inhibition of the direct reaction between the adsorbed oxygen and NH_3_ [121].

Ni(OH)_2_ has features such as non-toxicity, low cost, ease of synthesis, and semiconducting properties. Ni(OH)_2_/Ti_3_C_2_T*_x_* composites were synthesized via in situ electrostatic self-assembly. The sensor with ~7.8 wt% Ni(OH)_2_ revealed the highest response, of 13% ([(|R_g_ − R_a_|)/R_a_] × 100) to 50 ppm NH_3_ gas at RT. A further increase in Ni(OH)_2_ resulted in the partial aggregation of Ni(OH)_2_, causing a decrease in the number of adsorption sites and the sensing response. The formation of interfacial Schottky junctions between the two components and the increase in adsorption sites owing to the high surface area (54 m^2^/g) are attributed to the sensing mechanism [122].

A BiOCl-Ti_3_C_2_T*_x_* MXene composite with an NS morphology, excellent homogeneity, and good electronic characteristics was synthesized for sensing studies. It revealed a high response to 34.58 to 100 ppm NO_2_ gas at 80%RH. The high response of the gas sensor was attributed to the formation of p-p heterojunctions between BiOCl and MXene [123].

The SnS_2_/Ti_3_C_2_ MXene composites were produced via electrostatic interactions. The response to 50 ppm acetone was 29.8% ([(|R_g_ − R_a_|)/R_a_] × 100) at RT, and the t_res_ and t_rec_ were ~90 and 355 s, respectively. The oxygen-containing functional groups on Ti_3_C_2_ formed hydrogen bonds with acetone. Electrons flowed from Ti_3_C_2_T*_x_* to SnS_2_ to form heterojunctions with potential barriers, the heights of which were changed upon exposure to target gas. The sensor could detect acetone in both the optical and electrical modes. To demonstrate the optical mode of the sensor, the sensor signal was connected to an LED, and the blue light evolution images of the LED at various acetone concentrations were processed. The brightness of the LED increased with increasing acetone concentration (Figure 17a–d). A favorable correspondence was observed between the Euclidean distance and acetone concentration (1–100 ppm); the Euclidean distance was 109 for 50 ppm acetone [124].

Table 2 presents the RT gas-sensing properties of composite MXene-based gas sensors. They are mostly used for detection of NH_3_, methanol, NO_2_, HCHO, ethanol and acetone gases at RT.

#### 2.2.6. Doped/Decorated MXenes

Doping is a popular method for enhancing the gas-sensing properties of metal oxides [125]. Few studies on the doping of MXenes for gas-sensing applications have been reported. Generally, noble metals with catalytic activity are used for decoration on the sensing materials, and since they are much more expensive than other materials, fewer studies have been conducted using noble metal decoration on MXenes. Also, doped MXenes are less studied relative to composite-based MXenes for gas-sensing application due to the lower impact of doping on the gas response relative to heterojunctions. However, in future studies much more attention should be paid to doped and decorated MXenes for gas-sensing studies. Heteroatom additions to MXenes can go to lattice sites, functional group sites or become adsorbed on surfaces. In general, element doping effects are as follows: (i) generation of active species and increase in conductance; (ii) adjustment of the electronic structure by introduction of defects; (iii) changing of the surface nature and chemical bonds in MXene; (iv) adjustment of the surface chemical properties to increase catalytic performance [126]. In this regard, S atoms with high electronegativity can decrease the electron density of the Ti atom, leading to a higher binding energy than that of Ti-C bonds. In a relevant study, it was demonstrated that the S doping of Ti_3_C_2_T*_x_* MXene led to a higher gas-sensing response to toluene than that of the pristine sensor. An enhanced response of 214% ([(|R_g_ − R_a_|)/R_a_] × 100) to 1 ppm toluene was obtained after sulfur doping. Expansion of the interlayer spacing after sulfur doping has been reported; therefore, a larger surface area resulted in effective gas diffusion and provided more sites for toluene gas. Furthermore, the S at the surface of the MXenes acted like oxygen ions, leading to the expansion of the electron depletion layer (EDL) on the MXene. Upon interaction of toluene gas with these adsorbed sulfur species, they react with sulfur ions, and the liberated electrons increase the concentration of electrons, leading to the appearance of a sensor signal. Moreover, owing to the donating effect of the ethyl group, a remarkable enhancement in the activity of the H_2_ atoms on the benzene ring was observed, leading to enhanced selectivity for toluene gas. In addition, DFT calculations revealed an increase in the binding energy of toluene to the S-doped MXenes [127]. In another study, nitrogen-doped MXene (N-MXene) composited with poly 3, 4-ethylenedioxythio-phene (PEDOT) and poly (4-styrenesulfonate) (PSS) (PEDOT:PSS) was used for sensing studies. The sensor with twice the amount of N-MXene relative to the polymer component exhibited a response of 25% ([(|R_g_ − R_a_|)/R_a_] × 100) to 25 ppm NH_3_ gas at RT, which was higher than pristine counterparts. The doped N atoms with an electron-donor nature activated electron-transfer reactions and increased the number of sorption sites. Furthermore, the formation of n-p heterojunctions between the two components was ascribed to the sensing enhancement. In addition, the partial oxidation of MXene led to the formation of some TiO_2_ NPs, which not only widened the interlayer spacing of MXene but also limited the restacking of the MXene layer [128].

A Au-decorated *α*-Fe_2_O_3_/Ti_3_C_2_T*_x_* MXene nanocomposite disclosed a response of 16.9% ([(|R_g_ − R_a_|)/R_a_] × 100) to 1 ppm NH_3_ gas at RT, with fast t_res_ and t_rec_ of 3 and 2 s, respectively. The surface area *α*-Fe_2_O_3_/Ti_3_C_2_T*_x_* MXene nanocomposite was 57.34 m^2^/g, which provided sufficient adsorption sites for the NH_3_ gas molecules. In addition, the presence of surface groups on MXene and the formation of heterojunctions between MXene/Fe_2_O_3_ and Au/MXene and Au/Fe_2_O_3_ contributed to the sensing signal. Moreover, Au, which has a good catalytic activity, has a promising effect on the adsorption and dissociation of oxygen and NH_3_ gases from the surface of gas sensors. Based on DFT studies, the adsorption energy of NH_3_ on nanocomposites (−8.31 eV) was much larger than that other gases, reflecting the good selectivity of the sensor to NH_3_ gas [129].

The Au-In_2_O_3_/Ti_3_C_2_T*_x_* MXene composite was prepared by a chemical synthesis route, followed by electrostatic self-assembly. The sensor exhibited a response of 31% ([(|R_g_ − R_a_|)/R_a_] × 100) to 5 ppm HCHO at RT. Moreover, t_res_ and t_rec_ were extremely rapid (5/4 s). The enhanced sensing properties were related to the catalytic effect of the Au NPs, the formation of heterojunctions between different components (Figure 18a), and the high surface area (27.75 m^2^/g) of the composite. Due to differences between the work functions of Au, In_2_O_3_ and Ti_3_C_2_T*_x_* and in intimate contact, the electrons move from In_2_O_3_ to Au, forming Schottky barriers in interfaces between In_2_O_3_ and Au. Also, in interfaces between In_2_O_3_ and Ti_3_C_2_T*_x_*, electrons move from In_2_O_3_ to Ti_3_C_2_T*_x_* to equate the Fermi levels. Accordingly, in air, both Schottky barriers and heterojunction barriers form in interfaces. In a HCHO atmosphere, upon reaction of HCHO with adsorbed oxygen, the electrons return to the sensor surface and the heights of barriers change. This finally led to modulation of the electrical resistance in the HCHO environment. In addition, based on the DFT calculations, the adsorption energy of HCHO on the Au-In_2_O_3_/Ti_3_C_2_T*_x_* MXene composite was higher than that of the other sensors, confirming the high performance of the composite sensor for HCHO gas from an energy perspective (Figure 18b–d) [130].

Table 3 presents the RT gas-sensing properties of doped or decorated MXene-based gas sensors. They are mostly used for detection of NH_3_, C_7_H_8_, and HCHO gases at RT.

## 3. Conclusions and Outlooks

In this review, we have discussed the RT gas-sensing properties of MXene-based gas sensors. Pristine MXene gas sensors without any modification often exhibit poor performance; hence, their surfaces can be modified to increase the number of surface functional groups or add new functional groups. Therefore, the response and selectivity can be increased. Composite fabrication with other materials, such as metal oxides, TMDS, and CPs, is a very popular and promising strategy for enhancing the RT performance of gas sensors. In particular, because of the highly intrinsic sensing properties of metal oxides, their composites with MXenes have led to the realization of high-performance gas sensors that can work at RT. Composites based on MXenes-TMDs have high surface areas and abundant surface groups, both of which are beneficial for gas sensing. Furthermore, composites with CPs are highly sensitive to NH_3_ because of the high intrinsic sensitivities of both MXenes and CPs to this gas. Compared to MXene composites, less attention has been paid to doped MXenes, and more studies are needed in the future. Ternary composites are also promising for sensing applications; however, the optimization of all components is often difficult, and in this regard, more detailed studies are needed. In addition, the use of UV light to promote surface reactions and increase the number of active surface sites is a promising technique for enhancing the RT-sensing properties of MXene-based gas sensors.

From a performance point of view, generally, pristine MXene gas sensors suffer from low sensitivity, relatively poor selectivity and long t_res_ and t_rec_, while doped and decorated MXene gas sensors show higher sensitivity, better selectivity and shorter dynamics. Also, composite MXene-based gas sensors, especially those with metal oxides, exhibit the highest sensitivity, good selectivity and the shortest t_res_ and t_rec_.

Thus, RT MXene-based gas sensors can be used to fabricate flexible gas sensors for wearable applications. However, since they must work at RT, where the humidity is generally high, their sensing properties should be less affected by the presence of humidity. Thus, surface treatment is a good strategy to decrease the hydrophilicity of MXenes. Decoration with noble metals such as Au, Pt, and Pd can be a good strategy for enhancing the selectivity of MXene-based gas sensors towards different gases, which requires further exploration.

## Figures and Tables

**Figure 1 sensors-23-08829-f001:**
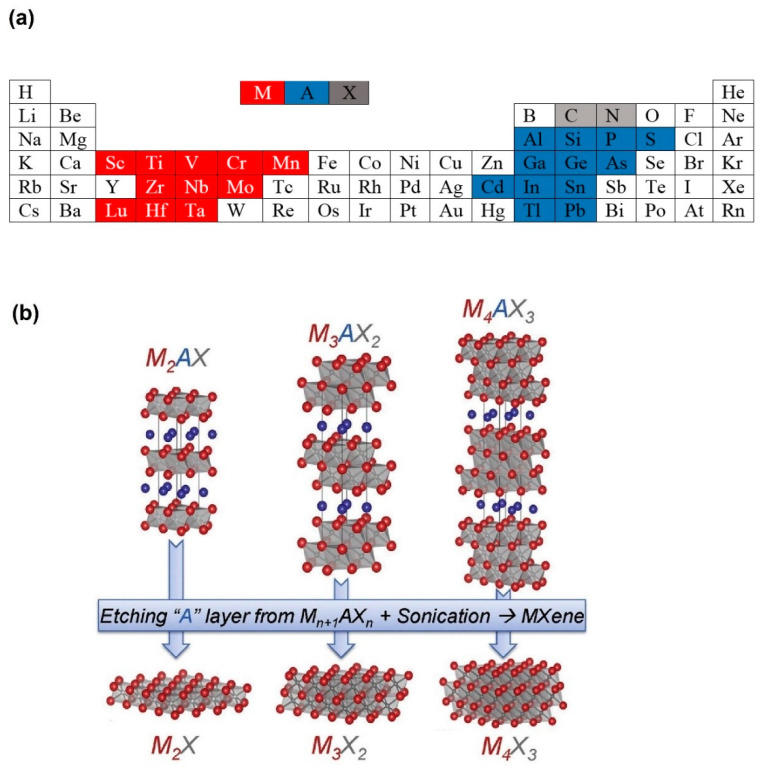
(**a**) Elements comprising MAX phase and (**b**) MXenes synthesized from MAX phases [38]. With permission from Elsevier.

**Figure 2 sensors-23-08829-f002:**
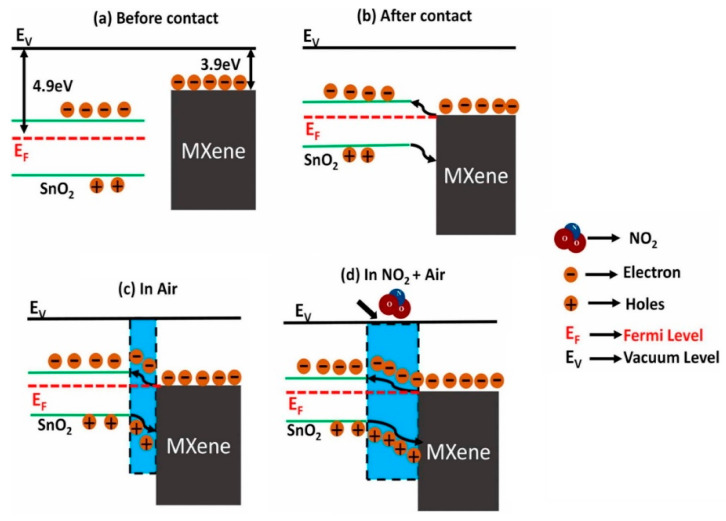
Energy band levels of MXene SnO_2_ (**a**) before and after contact in (**b**) vacuum (**c**) air (**d**) NO_2_ gas [76]. With permission from Elsevier.

**Figure 3 sensors-23-08829-f003:**
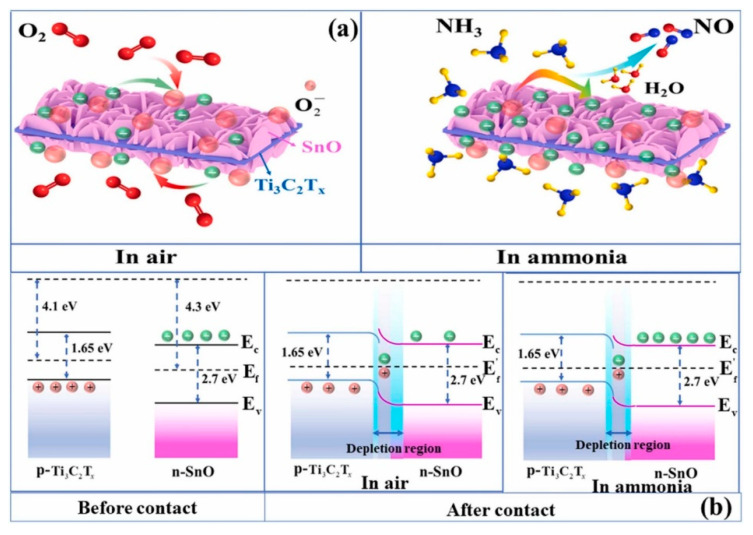
(**a**) Schematic of NH_3_-sensing mechanism and (**b**) the energy bands of Ti_3_C_2_T*_x_* MXene and SnO before contact, and after contact in air and NH_3_ [77]. With permission from Elsevier.

**Figure 4 sensors-23-08829-f004:**
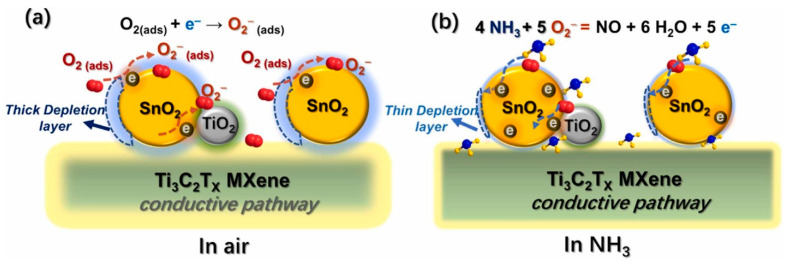
NH_3_-sensing mechanism of the Ti_3_C_2_T*_x_*-SnO_2_ composite in (**a**) air and (**b**) NH_3_ [78]. With permission from Elsevier.

**Figure 5 sensors-23-08829-f005:**
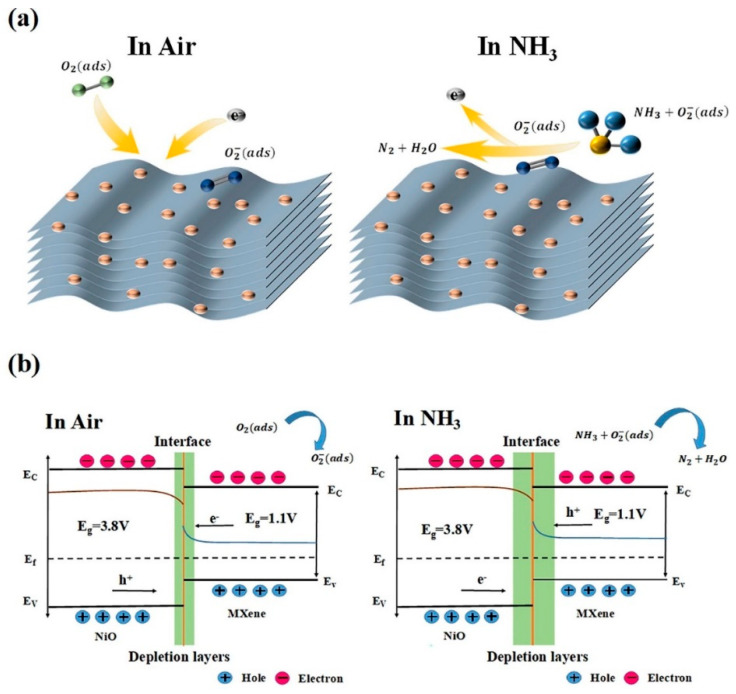
(**a**) Schematic image of the mechanism of NiO/Ti_3_C_2_T*_x_* Mxene sensor. (**b**) Formation of heterojunctions between NiO and MXene in air and acetone [79]. With permission from Elsevier.

**Figure 6 sensors-23-08829-f006:**
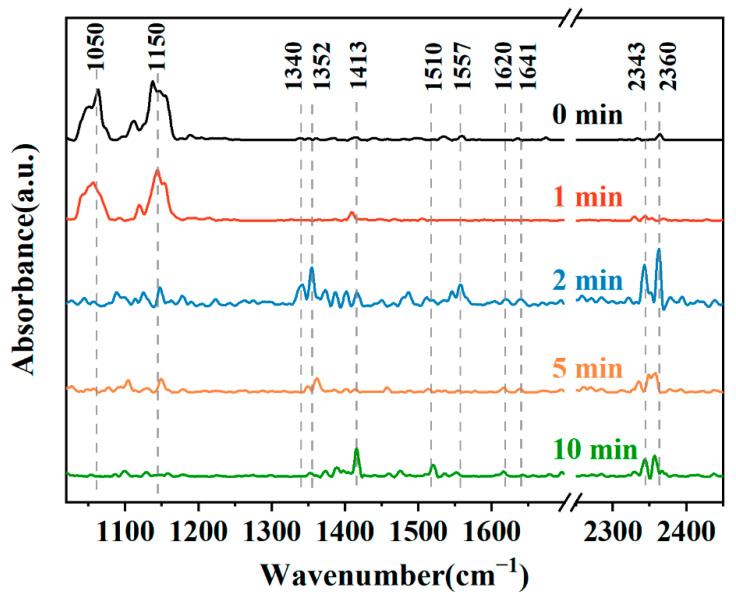
In situ FTIR spectra of the MXene/NiO sensor exposed to HCHO gas [80].

**Figure 7 sensors-23-08829-f007:**
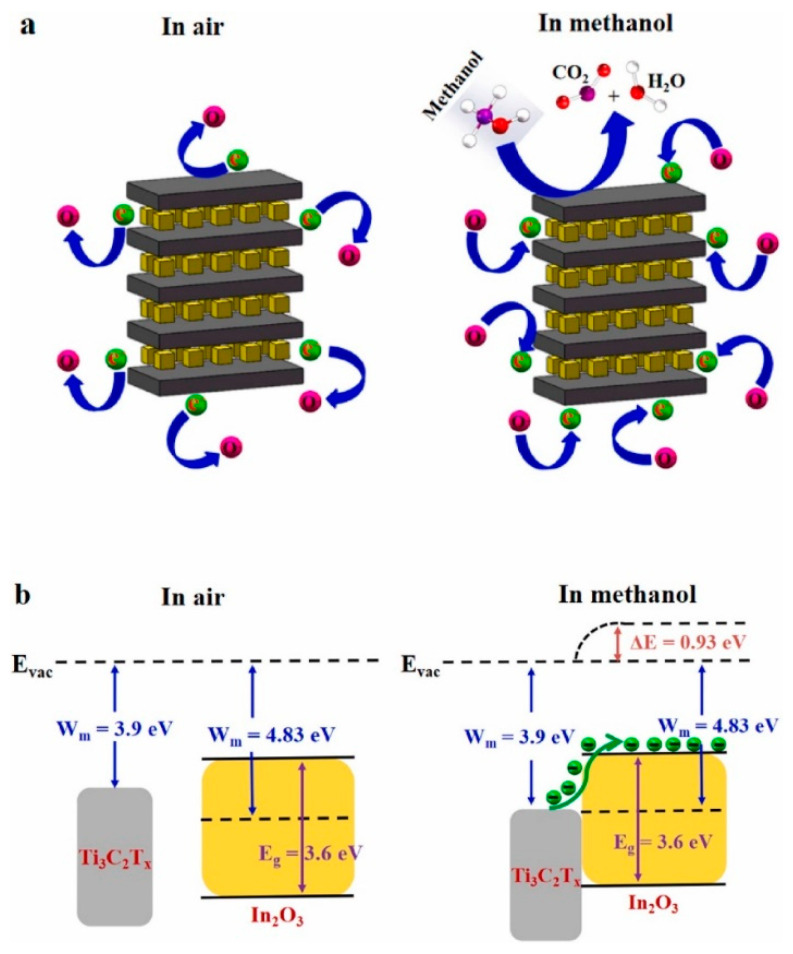
(**a**) Schematic of the sensing mechanism and (**b**) the energy band diagram of In_2_O_3_/Ti_3_C_2_T*_x_* MXene composites in air and methanol [83]. With permission from Elsevier.

**Figure 8 sensors-23-08829-f008:**
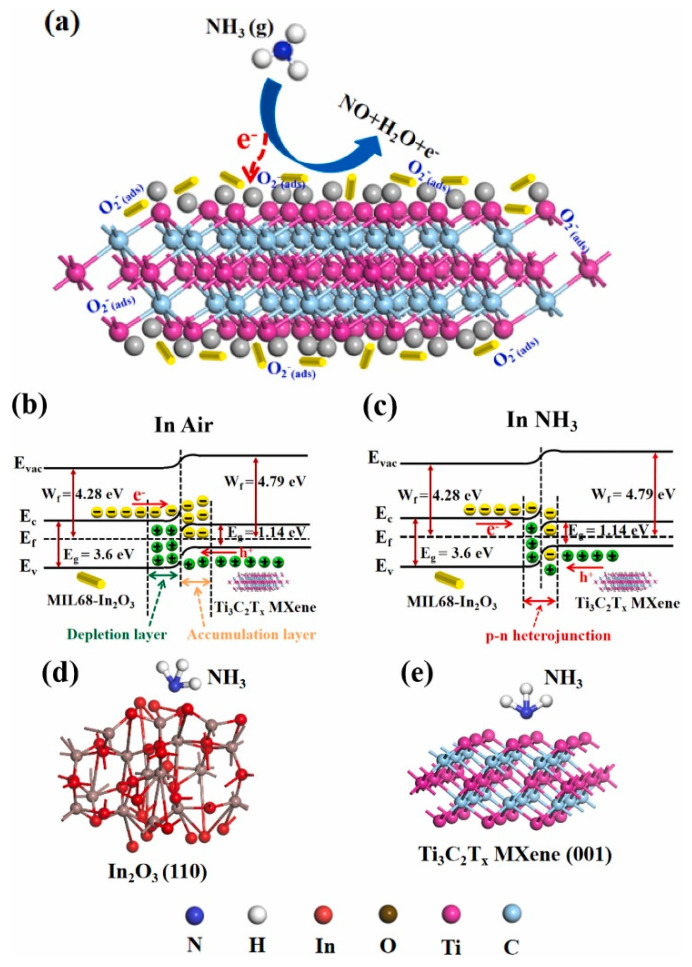
(**a**) Schematic of the mechanism of MOF-derived In_2_O_3_/Ti_3_C_2_T*_x_* MXene composites to NH_3_ (**b**,**c**) corresponding the energy bands in air and NH_3_ gas. (**d**,**e**) Optimized configurations of NH_3_ adsorption on In_2_O_3_ and Ti_3_C_2_T*_x_* MXene [84]. With permission from Elsevier.

**Figure 9 sensors-23-08829-f009:**
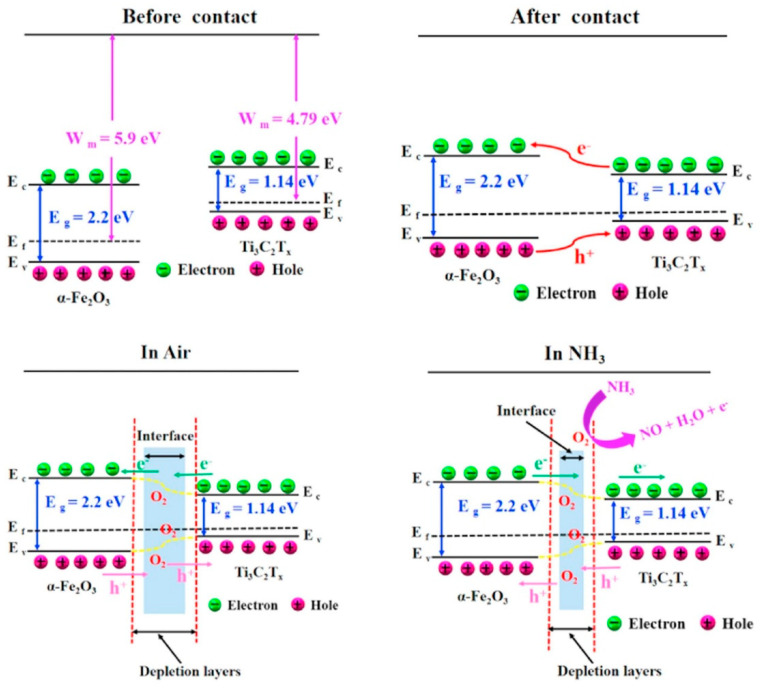
Energy bands of MXene and Fe_2_O_3_ before and after contact in the present of air and NH_3_ [86]. With permission from Elsevier.

**Figure 10 sensors-23-08829-f010:**
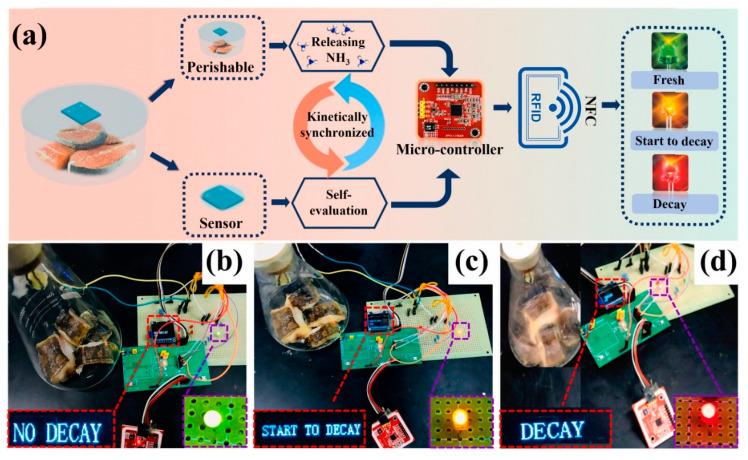
(**a**) Block diagram of the integrated circuit alarm system. Monitoring the fish status of (**b**) no decay, (**c**) start to decay, and (**d**) decay [97]. With permission from Elsevier.

**Figure 11 sensors-23-08829-f011:**
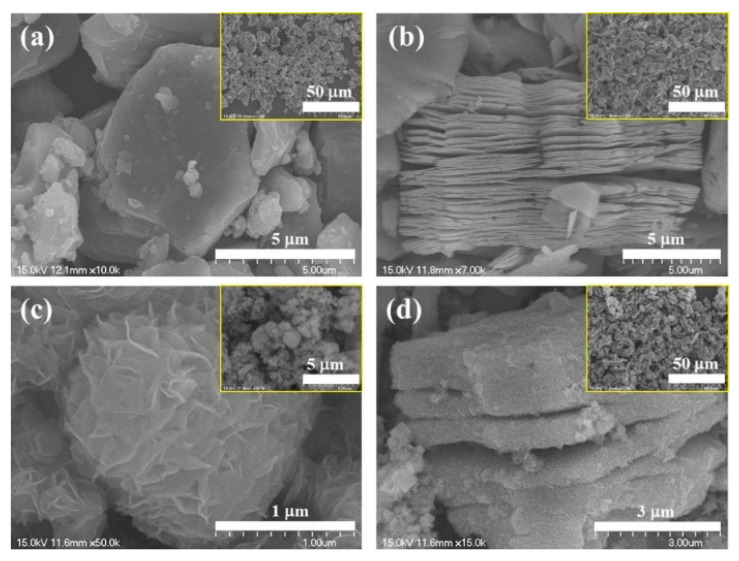
SEM images of (**a**) Ti_3_AlC_2_ powder, (**b**) Ti_3_C_2_T*_x_* MXene, (**c**) MoS_2_ nanoflower, and (**d**) MoS_2_/Ti_3_C_2_T*_x_* composite. Inset shows higher magnification images [103]. With permission from Elsevier.

**Figure 12 sensors-23-08829-f012:**
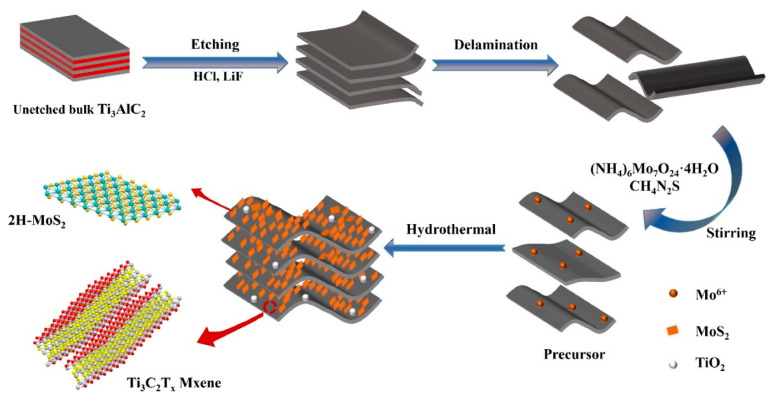
Schematic synthesis procedure of MoS_2_ NSs-Ti_3_C_2_T*_x_* MXene nanocomposite [104]. With permission from Elsevier.

**Figure 13 sensors-23-08829-f013:**
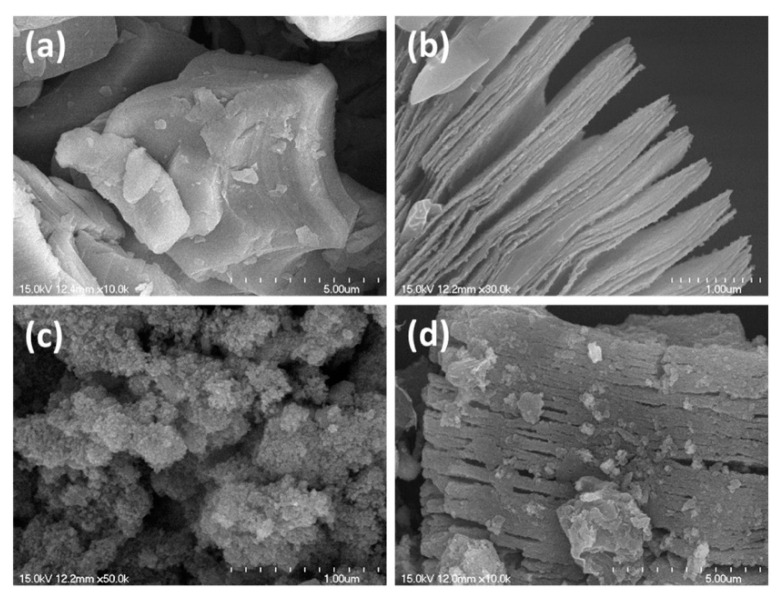
SEM images of (**a**) Ti_3_AlC_2_, (**b**) Ti_3_C_2_ MXene, (**c**) MoS_2_, and (**d**) Ti_3_C_2_-MoS_2_ [106]. With permission from Elsevier.

**Figure 14 sensors-23-08829-f014:**
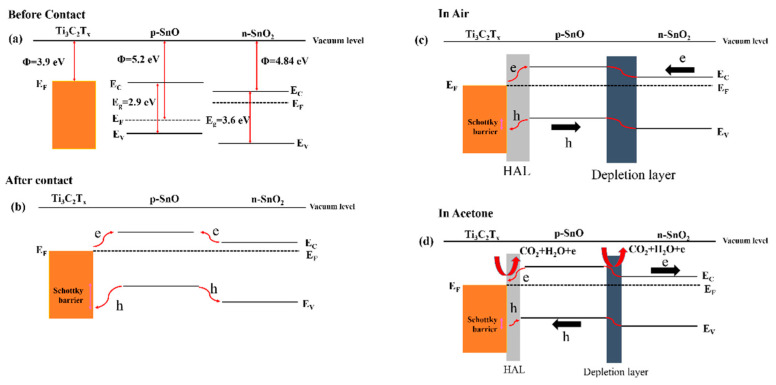
Schematic of the band structure in SnO-SnO_2_/Ti_3_C_2_T*_x_* nanocomposite (**a**) before and (**b**) after contact in (**c**) air and (**d**) acetone [115]. With permission from Elsevier.

**Figure 15 sensors-23-08829-f015:**
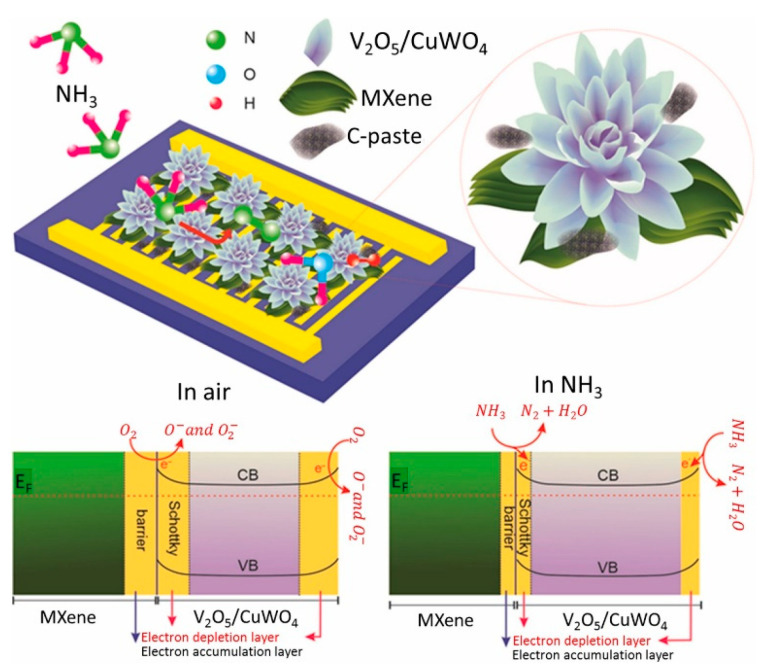
NH_3_-sensing mechanism of Ti_3_C_2_T*_x_*/V_2_O_5_/CuWO_4_ sensor [116]. With permission from Elsevier.

**Figure 16 sensors-23-08829-f016:**
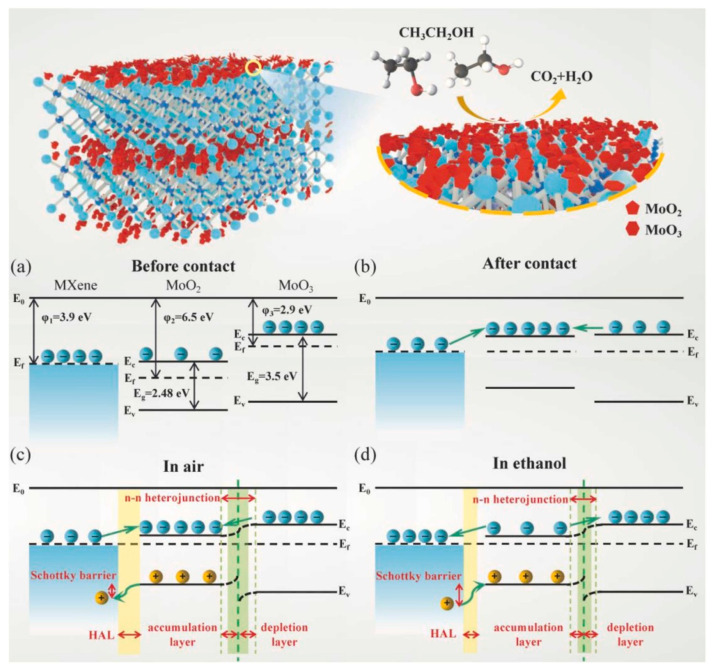
(**a**–**d**) Schematic of the gas-sensing mechanism of MoO_2_/MoO_3_/MXene-based sensor [118]. With permission from Elsevier.

**Figure 17 sensors-23-08829-f017:**
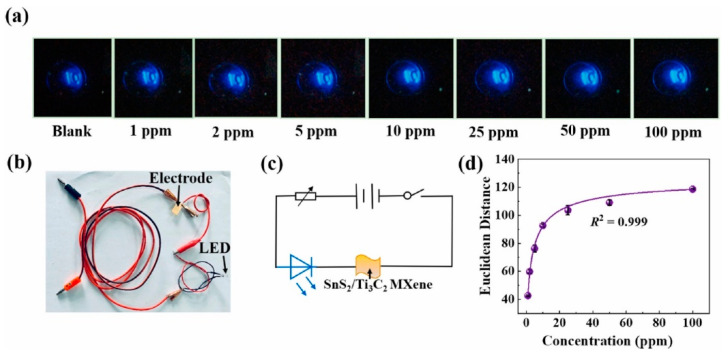
(**a**) Images and light evolutions of SnS_2_/Ti_3_C_2_ MXene-based visual sensor to detect acetone. (**b**) Photograph and (**c**) diagram of circuit. (**d**) The fitted curve of the Euclidean distance with acetone concentration [124]. With permission from Elsevier.

**Figure 18 sensors-23-08829-f018:**
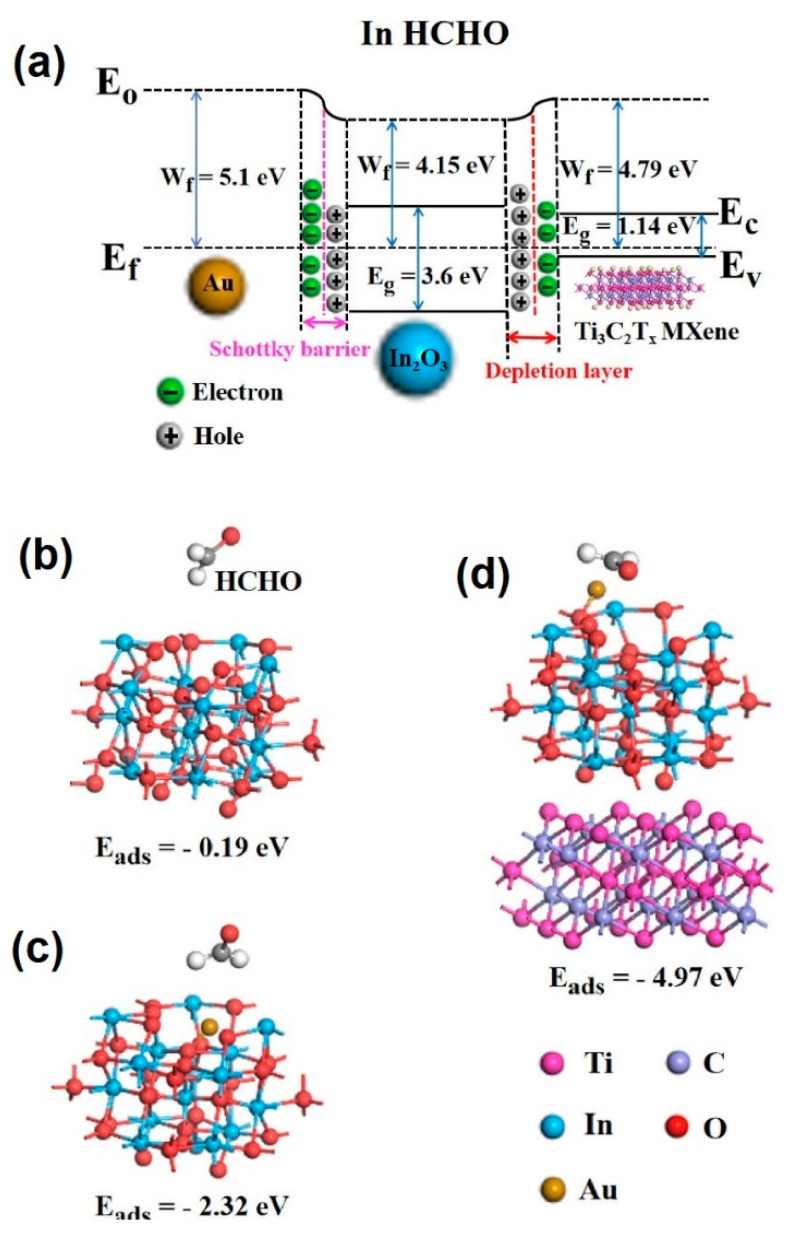
(**a**) The energy bands of Au-In_2_O_3_/Ti_3_C_2_T*_x_* MXene in HCHO. The optimized structure for HCHO adsorption system of (**b**) initial In_2_O_3_, (**c**) Au-In_2_O_3_ and (**d**) Au-In_2_O_3_/Ti_3_C_2_T*_x_* MXene composite [116]. With permission from Elsevier.

**Table 1 sensors-23-08829-t001:** RT gas-sensing properties of pristine MXene-based gas sensors.

Sensing Material	Gas	Conc. (ppm)	Response (%) ([(|R_g_ − R_a_|)/R_a_] × 100)	Ref.
Pristine Ti_3_C_2_T*_x_*	NH_3_	100	21%	[51]
Pristine Ti_3_C_2_T*_x_*	NH_3_	500	6	[56]
Plasma-exposed Ti_3_C_2_T*_x_*	NO_2_	10	13.8	[57]
Surface-modified Ti_3_C_2_T*_x_*	Ethanol	20	8	[59]
Surface-modified Ti_3_C_2_T*_x_*	NO_2_	25	26.8	[60]
Pristine Ti_3_C_2_T*_x_*	Acetone	100	100	[68]
Mo_2_CT*_x_*	CO_2_	50	2.3	[69]

**Table 2 sensors-23-08829-t002:** RT gas-sensing properties of composite MXene-based gas sensors.

Sensing Material	Gas	Conc. (ppm)	Response (%) ([(|R_g_ − R_a_|)/R_a_] × 100)	Ref.
SnO_2_/Ti_3_C_2_T*_x_* composite	NH_3_	40	40	[75]
Ti_3_C_2_T*_x_*-SnO_2_ composite	NH_3_	100	3.1	[78]
NiO/Ti_3_C_2_T*_x_* MXene nanocomposites	NH_3_	50	6.13	[79]
Ti_3_C_2_T*_x_* MXene (30–100 wt%)/In_2_O_3_ composite	NH_3_	20	100	[81]
In_2_O_3_ nanocubes/Ti_3_C_2_T*_x_* MXene composites	Methanol	5	29.6	[83]
Hollow In_2_O_3_ microbutes (2–5 µm) were attached on the exfoliated Ti_3_C_2_T*_x_* MXene	NH_3_	5	60.6	[84]
α-Fe_2_O_3_/Ti_3_C_2_T*_x_* MXene	Acetone	5	16.6	[85]
Rose-like α-Fe_2_O_3_/Ti_3_C_2_T*_x_* MXene composite	NH_3_	5	18.3	[86]
Ti_3_C_2_T*_x_*/WO_3_ nanocomposite	NO_2_	0.2	78	[87]
WO_3_—Ti_3_C_2_T*_x_* nanocomposite	NH_3_	1	22.3	[88]
Ti_3_C_2_T*_x_* MXene/CuO	NH_3_	5	46.7	[89]
Ti_3_C_2_T*_x_*/CuO nanocomposites	NO_2_	50	57	[90]
V_2_CT*_x_*/V_3_O_7_ nanocomposites	NO_2_	100	16	[91]
Ti_3_C_2_T*_x_* MXene/ZnO NRs	NO_2_	0.2	346	[94]
Ti_3_C_2_T*_x_*-ZnO nanocomposite	NO_2_	20	367.63	[96]
ZnSnO_3_/Ti_3_C_2_T*_x_* MXene composite	HCHO	100	194.7	[98]
Ti_3_C_2_T*_x_*-WSe_2_ composite	Ethanol	40	9	[101]
MoS_2_/Ti_3_C_2_T*_x_* composite	NH_3_	20	45	[103]
MoS_2_/Ti_3_C_2_T*_x_* composite	NO_2_	100	65.6	[104]
T_3_C_2_T*_x_*/WS_2_ nanocomposites	NO_2_	2	55.6	[105]
Ti_3_C_2_-MoS_2_ composite	NO_2_	10	35.8	[106]
PEDOT and PSS on Ti_3_C_2_T*_x_* MXene	NH_3_	100	36.6	[110]
PANI NPs Ti_3_C_2_T*_x_* NSs	Ethanol	100	27.4	[112]
CPAM/Ti_3_C_2_T*_x_* MXene composite	NH_3_	100	25	[116]
Ti_3_C_2_T*_x_* MXene@TiO_2_/MoS_2_ composite	NH_3_	100	164	[117]
Ti_3_C_2_T*_x_* MXene/rGO/SnO_2_ aerogel	HCHO	10	54.9	[120]
Ni(OH)_2_/Ti_3_C_2_T*_x_* composite	NH_3_	50	13	[122]
SnS_2_/Ti_3_C_2_T*_x_* MXene composite	Acetone	50	29.8	[124]

**Table 3 sensors-23-08829-t003:** RT gas-sensing properties of doped or decorated MXene-based gas sensors.

Sensing Material	Gas	Conc. (ppm)	Response (%) ([(|R_g_ − R_a_|)/R_a_] × 100)	Ref.
S-doped Ti_3_C_2_T*_x_* MXene	C_7_H_8_	1	214	[127]
N-doped Ti_3_C_2_T*_x_* MXene-PEDOT:PSS	NH_3_	25	25	[128]
Au-decorated *α*-Fe_2_O_3_/Ti_3_C_2_T*_x_* MXene	NH_3_	1	16.9	[129]
Au-decorated In_2_O_3_/Ti_3_C_2_T*_x_* MXene composite	HCHO	5	31	[130]

## Data Availability

These data can be found only in this article.

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
