# Peer review of "Room Temperature Chemiresistive Gas Sensors Based on 2D MXenes"

_sensors, 2023, doi:10.3390/s23218829_

Round 1

Reviewer 1 Report

Mirzaei et al. review room-temperature chemiresistive gas sensors based on MXenes, two-dimensional crystalline materials with the general formula M3X2Tx, where M is an early transition metal, X is carbon or nitrogen, and T is a surface termination, usually ‒O, ‒OH, or ‒F. MXenes have a number of properties that make them attractive for sensors, including high conductivity, large surface area, tunable bandgap, and mechanical flexibility. The authors describe sensors built from pristine MXenes, doped and decorated species, and composites with metal oxides, transition-metal dichalcogenides, and conducting polymers. The review is generally well-written and well-referenced but should be revised in several respects before being considered for publication.

1.      Other reviews of MXenes should be referenced and their different perspectives acknowledged; examples include reviews by Li et al. (Nat. Rev. Chem. 6 (2022) 389), Bhat et al. (npj 2D Mater. Appl. 5 (2021) 61), and Murali et al. (ACS Nano 16 (2022) 13370; Sustainable Energy Fuels 5 (2021) 5672).

2.      In Section 2, it would be helpful if the common terminations of MXenes were given.

3.      Sensor responses are usually given in terms of percentages (for example, lines 65-66). These figures presumably refer to changes in resistance, but the metric should be defined, as it is in the articles in the bibliography.

4.      Similarly, the discussion of preparation methods at the end of Section 2 should be significantly expanded. Otherwise, it is difficult to find context for statements like “… to study the effect of the Mo2CTx amount in the suspension, on the sensing performance to toluene gas, five Mo2CTx suspensions (0.011, 0.033, 0.066, 0.36, and 0.66 mg/mL) were prepared” (lines 170-172), or for references to hydrothermal, solvothermal, or template methods of synthesis.

5.      The figures with complex energy band diagrams should be discussed in much more detail.

6.      Some abbreviations (e.g., PEDOT, PSS) are defined well after they are first encountered, and some are never defined (e.g., PANI, HAL).

7.      The sentence in lines 606-608 is internally repetitive and needs to be edited.

Review punctuation and sentence structure; some responses lack percentage symbols.

Author Response

Reply is uploaded as a separate file.

Reviewer 2 Report

Review paper: ”Room Temperature Chemiresistive Gas Sensors Based on 2D MXenes“

The review paper is on the novel topic of a relatively new material class application in room temperature gas sensing. The review is not consistent and requires some improvements. Some remarks:

1.     The introduction is too short and does not detail the exclusiveness and advantages of RT gas sensors. This section should be extended as it looks like an abstract.

2.     2. MXenes: a brief introduction part is too focused on the structures of MXenes. However, the critical information for the readers is to present MXenes as a class of materials, explain the principal scheme of synthesis, and highlight that MXenes can be synthesized in multilayered and delaminated forms. These are the main factors essential to understand for the reader to have a basic understanding of the studies presented in this review.

3.     Moreover, MXenes, especially Ti3C2Tx, are mainly used in room temperature gas sensors because of MXenes' sensitivity and low thermal stability. It should be explained and highlighted in the section “2. MXenes: a brief introduction”. The oxidation and/or the state of MXenes oxidation can be investigated by Raman spectroscopy. Please find this paper: Chemosensors 2021, 9(8), 223; https://doi.org/10.3390/chemosensors9080223; it is explained in Figure 4, where oxide formation is assigned to band ~130 cm-1.

4.     More attention should be paid to the stability of MXenes and MXenes based gas sensors' sensitivity drop over time.

5.     The following sections should be split into multilayered and delaminated MXenes applications. Especially discussing bare MXenes applications.

6.     In the sections overviewing MXenes composites application in RT gas sensing, the influence of MXenes morphology should be highlighted.

7.     Review lack of tables to summarize data. It should be a table for the sensors' performance in every section.

8.     Sensitivity, selectivity, degradation, and recovery are essential parts of sensor performance. It should be discussed in the review, with the connection to different structures and morphology MXenes.

9.     As the stability and selectivity issues are not well covered in the current version of the manuscript, the need and logic behind the modification of MXenes are not fully explained. The stability and selectivity should be explained prior to discussing modified MXenes performances.

The text should be double-checked.

Author Response

Reply is uploaded as a separate file.

Reviewer 3 Report

In this review, the focus is on room temperature (RT) gas sensing properties of MXene-based gas sensors. The text is correctly categorized and provides a comprehensive overview of the current state of MXene-based gas sensors. With appropriate expansion of the introduction section, this manuscript can be considered for publication.

1-      The introduction section is written very short and it deprives the reader of the opportunity to communicate with the topic of the review paper:

-          Discuss more about challenges associated with gas sensing, particularly the need for sensors that can operate at room temperature. Mention how traditional gas sensors often require high temperatures for operation, which can be energy-intensive and limit their usability in certain applications.

-          Highlight 2D MXenes exceptional properties that make them attractive candidates, such as their ability to interact with gas molecules, making them sensitive to changes in gas concentration.

-          Clearly state the objectives of your review paper. Explain that your paper aims to provide a comprehensive overview of the recent advancements in the development and application of room temperature chemiresistive gas sensors based on 2D MXenes. Mention that you will cover the synthesis methods, sensing mechanisms, performance characteristics, and potential applications of these sensors.

-          Briefly outline the structure of the review paper, indicating how you will organize the content. Mention that you will start with an overview of MXenes, followed by a discussion of their synthesis methods, then delve into their applications in room temperature chemiresistive gas sensing, and conclude with a summary of current challenges and future prospects.

-          Emphasize the potential impact of 2D MXene-based room temperature gas sensors in advancing various fields and solving real-world problems.

-          Acknowledge any limitations in the scope of your review, such as the specific types of gases or applications covered, and clarify any potential bias in the selection of research papers or studies included.

-          You mentioned that MOX gas sensors operate at high temperatures, but there are reports that they can work at room temperature (Refer to DOI: 10.1088/1361-6528/acc6d7). Elaborate on the advantages and disadvantages of these room temperature MOX gas sensors, compare with 2D MXene-based room temperature.

2-      Briefly describe some common methods used for this etching process.

3-      Provide more and deeper details about the gas sensing mechanism of Pristine MXene gas sensors first of the section.

4-      Explain why achieving to low electrical noise and strong signals in gas sensors is difficult.

5-      Could you elaborate on how the numerous heterojunctions contribute to enhanced gas response?

6-      You mentioned: “few studies on the doping of MXenes for gas-sensing applications have been reported.”  What’s the challenge of decoration or doping?

7-      Can you point out as a conclusion which sensing parameters (Sensitivity, Stability, Selectivity, Response and Recovery time and ...) in any of the mentioned sensor types had the most impact?

Minor:

1-    In the table in Figure 1, dim the red color to highlight the text.

2-    Add reference for this sentence: “High-performance gas sensors should have low electrical noise owing to their high conductivity, and strong signals owing to their strong and abundant adsorption sites.”

3-    Check line 294.

4-    The textin figure 14 is unclear.

Author Response

Reply is uploaded as a separate file.

Round 2

Reviewer 1 Report

My comments have been adequately addressed.

A few minor spelling and grammatical errors.

Author Response

Thank you for the comments: My comments have been adequately addressed.

Reviewer 2 Report

Dear Authors,

thank you for the improvements. The manuscript is ready for publishing.

Author Response

Thank you for the comments: thank you for the improvements. The manuscript is ready for publishing.